# Verification and Training of Neural Networks for Robustness Against Neuron Pruning

## Abstract

Structured neuron pruning removes entire hidden units to reduce model size and computation, but often leads to unpredictable accuracy degradation. Existing pruning methods typically rely on heuristic importance scores and provide no formal guarantees on the behavior of pruned models. In this work, we propose a certifiable approach for structured neuron pruning in fully connected layers of feedforward neural networks that guarantees robustness against all pruning masks satisfying a given layer-wise sparsity budget. We further develop a computable upper bound on the worst-case change in pairwise class margins induced by neuron pruning. The analysis models pruning as row-zeroing (equivalently, neuron gating via binary masks) in the weight matrices and bounds the resulting deviation via operator-norm-based error propagation. These bounds are then used to develop a margin-aware robust training objective for certifiable pruning robustness. Experiments on MNIST and CIFAR-10 show that the resulting models achieve non-trivial certified accuracy under a range of pruning budgets and that our robust training substantially improves both certified and empirical robustness over standard baselines.

## 1 Introduction

Structured pruning is a widely used model-compression technique that removes entire neurons, channels, or filters to reduce computational cost and enable efficient deployment (Cheng et al., 2024; He & Xiao, 2023). Most existing methods remain heuristic-driven, relying on criteria such as weight magnitude, norms, loss estimates, or sparsity-inducing regularization (Cheng et al., 2024; Lahav & Katz, 2021; Huang & Wang, 2018; Jiang et al., 2023; Liu et al., 2020). Consequently, pruning performance can be sensitive to the chosen criterion, and most methods provide no formal guarantee on the resulting model accuracy, often relying on fine-tuning to recover performance.

While some works provide guarantees for pruning (Dong et al., 2017; Pitas et al., 2018; Ye et al., 2020; El Halabi et al., 2022), these guarantees are tied to specific pruning algorithms and do not generalize beyond the masks they produce. In some deployment settings, however, the realized pruning pattern is not fixed in advance. For example, dynamic inference methods may activate different sub-networks at runtime according to input-dependent computational demands or changing resource budgets (Bengio et al., 2015; Lin et al., 2017; Yu et al., 2018). Similarly, heterogeneous federated or on-device systems may deploy different pruning masks and sparsity levels across clients or devices (Zhou et al., 2022; Yi et al., 2024). Recent work further shows that critical feed-forward neurons can be deliberately pruned during inference to compromise model behavior (Wu et al., 2025). In these settings, guarantees tied to a particular pruning scheme do not automatically transfer when the pruning criterion, mask-selection policy, or deployment configuration changes. Generally, this requires a modified verification procedure and new analyses for each new setting. This motivates stronger mask-agnostic guarantees that are independent of the pruning pattern realized at deployment, as long as it remains within the prescribed sparsity budget.

In this work, we take a different perspective and view structured neuron pruning as a form of combinatorial model perturbation, where each pruning mask defines a distinct sub-network within an exponentially large family. Unlike prior work that focuses on identifying a single performant sub-network, we aim to provide

*worst-case, mask-agnostic* guarantees over all sub-networks satisfying a layer-wise sparsity budget. This perspective connects model compression with certified robustness, enabling guarantees under discrete structural perturbations arising from algorithmic choices, system constraints, or hardware faults.

Formally, we consider multiclass classification with feedforward networks. Given a trained network $f$ and a layer-wise pruning budget $S$, which specifies the maximum number of neurons that may be removed in each fully connected layer, we define $\mathcal{F}_{\text{pruned}}$ as the family of all pruned subnetworks of $f$ that satisfy $S$. The verification problem is to verify that every $\hat{f} \in \mathcal{F}_{\text{pruned}}$ preserves the correct prediction for a given input. Since this family grows combinatorially with network size, exhaustive evaluation is intractable. Leveraging advances in non-convex global optimization for deep learning (Tjeng et al., 2017; Wong & Kolter, 2018; Chiu & Zhang, 2023), we reformulate pruning robustness as a *mixed-integer linear program* (MILP), solvable by standard solvers such as Gurobi (Cheng & Li, 2022).

To enable training-time robustness, we further derive a computable upper bound on the worst-case deviation in pairwise class margins between the original network and any admissible pruned sub-network. Based on this analysis, we propose a margin-aware, bound-guided training objective that improves robustness to pruning.

We evaluate our methods on different architectures trained on the MNIST and CIFAR-10 datasets. We show that our verification techniques based on MILP and worst-case margin deviation yield non-vacuous bounds across a range of sparsity budgets. Moreover, our robust training substantially improves the verified accuracy and empirical performance compared to standard baselines. In summary, our main contributions are as follows:

- We formalize robustness against structured neuron pruning in FC layers as a worst-case guarantee over a family of sub-networks defined by a layer-wise sparsity budget and develop a MILP-based verification method (Section 4.1).

- We derive computable upper bounds on worst-case pairwise margin deviation under pruning for single-layer, multi-layer, and all-layer settings (Section 4.2).

- We propose a theory-driven training objective that leverages our margin-based bounds to improve robustness to pruning under a given layer-wise sparsity budget (Section 4.3). We observe improvements in both certified and empirical robustness over standard baselines.

## 2 Related Work

There are a number of works that explored pruning techniques with performance guarantees (Lahav & Katz, 2021; Li et al., 2021; Gokulanathan et al., 2020; Aghasi et al., 2017; Dong et al., 2017; Pitas et al., 2018; Ye et al., 2020; El Halabi et al., 2022). For instance, Lahav & Katz (2021) and Gokulanathan et al. (2020) use verification techniques to identify redundant neurons whose removal provably preserves network outputs. Similarly, Aghasi et al. (2017) formulated an optimization problem to prune a network while keeping the pruned model's activations consistent with the original model on the training data. While these works demonstrate that formal guarantees in pruning are possible, their guarantees apply only to the particular masks produced by the corresponding selection strategies, such as layer-wise optimal brain surgeon (Aghasi et al., 2017), difference-of-convex-based pruning (Pitas et al., 2018), or submodular greedy selection (El Halabi et al., 2022). Consequently, no guarantee holds when the network is pruned using other techniques, and in such cases, a new verification routine must be developed and executed. In contrast, we provide guarantees that hold across all possible pruned models satisfying a given layer-wise sparsity budget.

Another line of work aims to prepare neural networks during training so that they are more amenable to post-hoc pruning. One approach focuses on shaping the loss landscape to improve robustness to parameter perturbations. In particular, flatness-based methods, including sharpness-aware minimization, encourage solutions that are less sensitive to weight changes, thereby improving post-pruning performance (Peste et al., 2022; Bair et al., 2023; Lee et al., 2025; Na et al., 2022). Complementary approaches explicitly promote sparsity or structural adaptability during training. For example, Khan & Stavness (2020) propose sparsity-inducing regularization schemes, while Gomez et al. (2019) introduce targeted dropout to encourage

robustness to neuron removal. Related methods, such as slimmable networks and once-for-all models, train shared weights that can be adapted to multiple subnetworks at different widths or sparsity levels (Yu et al., 2018; Cai et al., 2019). While these methods improve empirical performance across different pruning configurations, their evaluation is typically limited to a small subset of subnetworks or specific pruning strategies. As a result, they do not assess worst-case behavior and provide no formal guarantees on model robustness under arbitrary admissible pruning masks.

Our work is also closely related to certified robustness, which aims to provide guarantees on model predictions under bounded perturbations. Existing methods primarily focus on continuous perturbations, including adversarial input perturbations and weight perturbations (Hein & Andriushchenko, 2017; Wong & Kolter, 2018; Weng et al., 2018; Xu et al., 2020; Tsai et al., 2021a; Dang et al., 2025). However, these approaches do not capture the discrete, combinatorial nature of pruning, where perturbations are constrained binary masks over network structure. In contrast, we study structured neuron pruning as a form of combinatorial model perturbation under layer-wise sparsity constraints and enable worst-case guarantees over the resulting family of subnetworks. This setting differs fundamentally from continuous robustness and introduces unique challenges due to its combinatorial nature.

## 3 Problem Formulation

In this section, we present the notations used throughout the paper and define pruned models under a sparsity budget. We then formalize the two main problems addressed in this work: pruning-robust verification and pruning-robust training.

**Notations.** Sets and spaces are denoted by capital letters, except for $K$, the number of hidden layers in a neural network (NN), and $C$, the number of classes. For any positive integer $N$, we denote $[N] = \{1, \ldots, N\}$. Matrices are denoted by bold uppercase letters (e.g., $\mathbf{W}$), and vectors by bold lowercase letters (e.g., $\mathbf{x}$). We use $\mathbf{W}[i]$ to denote the $i$-th row of a matrix $\mathbf{W}$ and $\mathbf{x}[i]$ to denote the $i$-th entry of a vector $\mathbf{x}$. We use $\|\cdot\|_2$ to denote the Euclidean norm for vectors and the induced spectral norm for matrices, i.e., $\|\mathbf{W}\|_2 = \sup_{\|\mathbf{x}\|_2=1} \|\mathbf{W}\mathbf{x}\|_2$, and $\|\cdot\|_0$ to denote the cardinality of a vector's support.

**Model.** We consider a multiclass classification problem with $C$ classes, where pruning is applied to the fully connected (FC) layers of a neural network. To isolate the effect of neuron-level pruning on these layers, we model the network as a fully connected feedforward network with $K$ hidden layers. Let $\mathbf{x} \in \mathcal{X} \subseteq \mathbb{R}^{d_0}$ denote the input, and define the forward pass as

$$
\begin{aligned}
\mathbf{x}_0 &= \mathbf{x}, \\
\mathbf{z}_k &= \mathbf{W}_k \mathbf{x}_{k-1}, && k = 1, \ldots, K, \\
\mathbf{x}_k &= \sigma(\mathbf{z}_k), && k = 1, \ldots, K, \\
\mathbf{f}(\mathbf{x}) &= \mathbf{W}_{K+1} \mathbf{x}_K,
\end{aligned}
$$

where $\mathbf{W}_k \in \mathbb{R}^{d_k \times d_{k-1}}$, $k \in [K]$ are the hidden-layer weight matrices, and $\mathbf{W}_{K+1} \in \mathbb{R}^{C \times d_K}$ is the output-layer weight matrix, and $\mathbf{f}(\mathbf{x})$ denotes the vector of pre-softmax logits. The predicted class is $\hat{y}(\mathbf{x}) = \arg\max_{i \in [C]} f_i(\mathbf{x})$. Since the softmax transformation preserves the ordering of logits, it is omitted from the analysis. For simplicity, we also omit bias terms. This does not restrict generality, as biases can be incorporated by augmenting the input with a constant dimension.

### 3.1 Sparsity Budgets and Pruned Models

We define structured neuron pruning via layer-wise sparsity budgets $S = \{s_1, \ldots, s_K\}$, where $s_k$ specifies the maximum number of neurons that can be removed at hidden layer $k$ ($0 \le s_k \le d_k - 1$).

For each layer $k \in [K]$, let $M_k = \{\mathbf{m}_k \in \{0,1\}^{d_k} : \|\mathbf{1} - \mathbf{m}_k\|_0 \le s_k\}$ denote the set of admissible pruning masks, where $\mathbf{m}_k[i] = 0$ indicates that neuron $i$ is pruned. Given $\mathbf{m}_k \in M_k$, we define the pruning operator $\mathcal{P}_{\mathbf{m}_k}(\mathbf{x}_k) = \mathbf{m}_k \circ \mathbf{x}_k$, where $\circ$ denotes element-wise multiplication.

A pruned network $\hat{f}$ associated with a budget $S$ and masks $\{\mathbf{m}_k\}_{k=1}^K$ is defined recursively as

$$\hat{\mathbf{x}}_0^p = \mathbf{x}, \tag{1}$$

$$\hat{\mathbf{z}}_k = \mathbf{W}_k \hat{\mathbf{x}}_{k-1}^p, \tag{2}$$

$$\hat{\mathbf{x}}_k = \sigma(\hat{\mathbf{z}}_k), \tag{3}$$

$$\hat{\mathbf{x}}_k^p = \mathcal{P}_{\mathbf{m}_k}(\hat{\mathbf{x}}_k), \qquad k = 1, \ldots, K. \tag{4}$$

subject to $\mathbf{m}_k \in M_k$. The final output is given by $\hat{f}(\mathbf{x}) = \mathbf{W}_{K+1} \hat{\mathbf{x}}_K^p$.

**Remark 1** *The operator $\mathcal{P}_{\mathbf{m}_k}$ models neuron removal by zeroing out the corresponding activations. This is equivalent to removing the associated rows and columns in the weight matrices during forward propagation if activation functions satisfy $\sigma(0) = 0$, ensuring that pruned neurons do not contribute to subsequent layers.*

### 3.2 Verification and Training for Pruning Robustness

We consider two problems in this work. First, we aim to provide robustness guarantees for all pruned models satisfying a given sparsity budget. Second, we tackle the problem of training a NN such that it is robust against such pruning.

**Problem 1 (Pruning-robust verification)** *Let $f$ be a neural network with $K$ hidden fully connected layers and a layer-wise sparsity budget $S = \{s_1, \ldots, s_K\}$, where $0 \leq s_k \leq d_k - 1$. Let $M_{\text{pruned}} = \{\mathbf{m} = \{\mathbf{m}_1, \ldots, \mathbf{m}_K\} : \|\mathbf{1} - \mathbf{m}_k\|_0 \leq s_k, \forall k \in [K]\}$ denote the set of admissible pruning masks. For a mask $\mathbf{m} \in M_{\text{pruned}}$, let $\hat{f}_{\mathbf{m}}$ denote the corresponding pruned network, and let $\hat{f}_{\mathbf{m}}^{(i)}(\mathbf{x})$ denote the logit of class $i \in [C]$. Given an input $\mathbf{x} \in \mathcal{X} \subseteq \mathbb{R}^{d_0}$ with true label $t$, the goal is to solve*

$$\min_{\mathbf{m} \in M_{\text{pruned}}} \gamma_{\mathbf{m}}(\mathbf{x}, t) = \min_{\mathbf{m} \in M_{\text{pruned}}} \min_{i \in [C], \, i \neq t} \left( \hat{f}_{\mathbf{m}}^{(t)}(\mathbf{x}) - \hat{f}_{\mathbf{m}}^{(i)}(\mathbf{x}) \right), \tag{5}$$

*which corresponds to the worst-case pairwise margin over all admissible pruning masks.*

In the case of classification, if the optimal value of equation 5 is strictly positive, then the prediction remains unchanged and correct under all admissible pruning masks. In this case, the model is said to be *pruning-robust* at input $\mathbf{x}$.

Reasoning over the set of admissible pruning masks involves a combinatorial discrete space, rendering Problem 1 NP-hard in general. Instead of solving it exactly, in this work, we develop tractable formulations that compute certified lower bounds on the worst-case margin.

To train a neural network that is robust against pruning under a given sparsity budget, we incorporate pruning-induced robustness into the training objective. Motivated by prior work on margin-based robustness and weight perturbations (Hein & Andriushchenko, 2017; Tsai et al., 2021a;b), we use the worst-case pruning-induced margin deviation as a training signal.

**Problem 2 (Pruning-robust training)** *Let $f$ be a neural network with parameters $\mathbf{W}$, a training set $\mathcal{D} = \{(\mathbf{x}_i, y_i)\}_{i=1}^{|\mathcal{D}|}$, and a layer-wise sparsity budget $S = \{s_1, \ldots, s_K\}$. For an input $\mathbf{x} \in \mathcal{X} \subseteq \mathbb{R}^{d_0}$ with true label $t$, and for any admissible pruning mask $\mathbf{m} \in M_{\text{pruned}}$, let $\hat{f}_{\mathbf{m}}$ denote the corresponding pruned network. Define the worst-case pairwise class-margin deviation as*

$$\Delta(S, \mathbf{x}, t) = \max_{\mathbf{m} \in M_{\text{pruned}}} \max_{i \in [C], \, i \neq t} \left| \left( \hat{f}_{\mathbf{m}}^{(t)}(\mathbf{x}) - \hat{f}_{\mathbf{m}}^{(i)}(\mathbf{x}) \right) - \left( f^{(t)}(\mathbf{x}) - f^{(i)}(\mathbf{x}) \right) \right|. \tag{6}$$

*The objective is to encourage $\Delta(S, \mathbf{x}, t)$ to remain small relative to the nominal margin $\gamma(\mathbf{x}, t) = f^{(t)}(\mathbf{x}) - \max_{i \neq t} f^{(i)}(\mathbf{x})$.*

*Indeed, for any admissible pruning mask $\mathbf{m}$, define $\gamma(\mathbf{x}, t, i) = f^{(t)}(\mathbf{x}) - f^{(i)}(\mathbf{x})$ and $\hat{\gamma}_{\mathbf{m}}(\mathbf{x}, t, i) = \hat{f}_{\mathbf{m}}^{(t)}(\mathbf{x}) - \hat{f}_{\mathbf{m}}^{(i)}(\mathbf{x})$ as the pairwise margins between the true class $t$ and class $i \neq t$ before and after pruning, respectively. The nominal margin satisfies $\gamma(\mathbf{x}, t) = \min_{i \neq t} \gamma(\mathbf{x}, t, i)$.*

Since each pairwise margin changes by at most $\Delta(S, \mathbf{x}, t)$, for every $\mathbf{m} \in M_{\text{pruned}}$ and $i \neq t$,

$$\hat{\gamma}_{\mathbf{m}}(\mathbf{x}, t, i) \geq \gamma(\mathbf{x}, t, i) - \Delta(S, \mathbf{x}, t).$$

Therefore, if $\Delta(S, \mathbf{x}, t) < \gamma(\mathbf{x}, t)$, then $\hat{\gamma}_{\mathbf{m}}(\mathbf{x}, t, i) > 0$ for every admissible pruning mask and every class $i \neq t$, which is sufficient to preserve the prediction. This observation motivates the pruning-robust training objective stated above.

**Remark 2** *The definition of $\Delta(S, \mathbf{x}, t)$ uses an absolute value and therefore captures a two-sided notion of pruning-induced margin deviation. This yields a sound but potentially conservative sufficient condition for certification, as margin increases are treated as deviations even though they do not compromise correct classification under pruning.*

The maximization in equation 6 over the discrete mask space $M_{\text{pruned}}$ is highly non-linear and non-differentiable. Instead, we derive tractable upper bounds on its optimal value, denoted by $\delta(S, \mathbf{x}, t)$. These bounds serve as surrogates of the worst-case deviation and enable the design of margin-aware, bound-guided training objectives for pruning-robust neural networks.

## 4 Methodology

In this section, we first show how to certify pruning robustness under all possible pruning masks via global optimization (Section 4.1). We then derive analytical upper bounds on the worst-case deviation of pairwise class margin induced by pruning at a given sparsity budget in three cases: single-layer pruning, all-layer pruning, and multi-layer pruning (Section 4.2). Finally, we leverage these bounds to design a surrogate training objective for pruning-robust learning (Section 4.3).

### 4.1 Mixed-Integer Linear Programming for Pruning-Robust Verification

Directly solving Problem 1 requires enumerating all admissible pruning masks, which is infeasible due to the combinatorial size of $M_{\text{pruned}}$. We therefore formulate the problem as a mixed-integer linear program (MILP) that jointly optimizes over all admissible pruning masks and their corresponding network activations, without explicitly enumerating the masks.

The verification objective remains the same as in Problem 1. The reformulation lies in the representation of its feasible set. Specifically, the network forward pass, activation functions, pruning masks, and sparsity constraints are represented within the mixed-integer linear formulation. Provided that valid finite bounds on all pre-activations and activations are available, solving the resulting formulation to global optimality provides a certified lower bound on the worst-case class margin over all admissible pruning masks. The corresponding constraints are described below.

**(1) Activation constraints.** For each layer $k \in [K]$ and neuron $i \in [d_k]$, let $\underline{\hat{z}}_k[i]$ and $\overline{\hat{z}}_k[i]$ denote valid finite bounds on the pre-activation $\hat{z}_k[i]$. For ReLU activations, we introduce binary variables $a_{k,i} \in \{0, 1\}$ and exactly encode all feasible ReLU activation patterns (Tjeng et al., 2017):

$$\hat{x}_k[i] \geq \hat{z}_k[i], \quad \hat{x}_k[i] \geq 0, \quad \hat{x}_k[i] \leq \overline{\hat{z}}_k[i] \, a_{k,i}, \quad \hat{x}_k[i] \leq \hat{z}_k[i] - \underline{\hat{z}}_k[i](1 - a_{k,i}). \tag{7}$$

For general activation functions, we instead use affine upper and lower bounds over the interval $[\underline{\hat{z}}_k[i], \overline{\hat{z}}_k[i]]$ (Zhang et al., 2018).

**(2) Pruning constraints.** For each layer $k \in [K]$ and neuron $i \in [d_k]$, neuron pruning is modeled as a binary gating operation $\hat{x}_k^p[i] = m_k[i]\hat{x}_k[i]$, where $m_k[i] \in \{0, 1\}$ indicates whether neuron $i$ is retained. Given valid bounds $L_k[i] \leq \hat{x}_k[i] \leq U_k[i]$, the bilinear terms are linearized exactly using McCormick envelopes (McCormick, 1976):

$$\begin{aligned}
\hat{x}_k^p[i] &\leq U_k[i]m_k[i], \\
\hat{x}_k^p[i] &\geq L_k[i]m_k[i], \\
\hat{x}_k^p[i] &\leq \hat{x}_k[i] - L_k[i](1 - m_k[i]), \\
\hat{x}_k^p[i] &\geq \hat{x}_k[i] - U_k[i](1 - m_k[i]).
\end{aligned} \tag{8}$$

The sparsity budget is enforced via: $\sum_{i=1}^{d_k} m_k[i] \geq d_k - s_k$.

**(3) Bound propagation.** We obtain valid finite bounds on all pre-activations and activations using interval bound propagation (IBP) from a bounded input domain $\mathcal{X} \subseteq \mathbb{R}^{d_0}$ (Gowal et al., 2018). For each layer $k \in [K]$ and neuron $i \in [d_k]$, given bounds $L_{k-1}[r] \leq \hat{x}_{k-1}^p[r] \leq U_{k-1}[r]$, the lower and upper bounds on the pre-activation $\hat{z}_k[i]$, denoted by $\underline{\hat{z}}_k[i]$ and $\overline{\hat{z}}_k[i]$, respectively, are computed via interval arithmetic:

$$\underline{\hat{z}}_k[i] = \sum_r \left( W_k[i,r]^+ L_{k-1}[r] + W_k[i,r]^- U_{k-1}[r] \right), \quad \overline{\hat{z}}_k[i] = \sum_r \left( W_k[i,r]^+ U_{k-1}[r] + W_k[i,r]^- L_{k-1}[r] \right). \quad (9)$$

The activation bounds $[L_k[i], U_k[i]]$ are then obtained by applying $\sigma$ over the interval $[\underline{\hat{z}}_k[i], \overline{\hat{z}}_k[i]]$.

**Remark 3** *The interval bounds obtained from IBP are used only to provide valid finite bounds for the mixed-integer encoding. Tighter bounds can improve computational efficiency but are not required for exactness. Since the mask variables are binary, the mask-activation products in equation 8 are represented exactly by the McCormick constraints. Thus, the exactness of the overall formulation depends on the activation encoding.*

*For ReLU networks, the constraints in equation 7 exactly represent all feasible activation phases. Therefore, provided that all pre-activation and activation bounds are valid and finite, the binary variables are enforced as integral, and the MILP is solved to global optimality, its optimal value equals the exact worst-case class margin over all admissible pruning masks. For non-ReLU activations represented using valid affine lower and upper bounds, the formulation is instead a relaxation, and its optimal value provides a sound lower bound on the true worst-case class margin.*

The scalability of the MILP formulation is primarily limited by the number of binary variables, which grows with network size. Each neuron introduces binary variables for both activation phases and pruning masks, leading to exponential worst-case complexity. In multi-class settings, the verification is typically decomposed into $C-1$ sub-problems (one per competing class), further multiplying the computational cost. These factors make the approach challenging for larger networks and complex datasets.

## 4.2 Upper bound on the worst-case margin deviation for robust training objective

**Assumptions for bound analysis.** For the derivation of margin-based deviation bounds, we assume that the activation function $\sigma$ is 1-Lipschitz and satisfies $\sigma(0) = 0$. Note that these conditions are not required for the MILP-based verification framework. Formally, let $\mathbf{m}_k \in \{0,1\}^{d_k}$ and define the masked weight matrix: $\mathcal{P}_{\mathbf{m}_k}(\mathbf{W}_k) := \mathrm{Diag}(\mathbf{m}_k)\mathbf{W}_k$. Under this assumption $\sigma(0) = 0$, pruning can be modeled as row masking, the pruned network $\hat{f}$ associated with masks $\{\mathbf{m}_k\}_{k=1}^K$ is defined recursively as

$$\hat{\mathbf{x}}_1^p = \sigma\left(\mathcal{P}_{\mathbf{m}_1}(\mathbf{W}_1)\mathbf{x}_0\right),$$
$$\hat{\mathbf{x}}_k^p = \sigma\left(\mathcal{P}_{\mathbf{m}_k}(\mathbf{W}_k)\hat{\mathbf{x}}_{k-1}^p\right), \quad k = 2, \ldots, K,$$
$$\hat{f}(\mathbf{x}) = \mathbf{W}_{K+1}\hat{\mathbf{x}}_K^p.$$

For any $\mathbf{m} \in M_{\mathrm{pruned}}$, define $\Delta\mathbf{x}_k(\mathbf{m}) = \hat{\mathbf{x}}_k^p - \mathbf{x}_k, k \in [K]$, as the layer-wise output deviation induced by $\mathbf{m}$, which accumulates pruning-induced perturbations from all preceding layers. For notational simplicity, we write $\Delta\mathbf{x}_k$ when its dependence on $\mathbf{m}$ is clear from context.

**Proposition 1 (Worst-case margin deviation)** *Let $f$ be a neural network with $K$ hidden fully connected layers and sparsity budget $S = \{s_1, \ldots, s_K\}$, which defines the set of admissible pruning masks $M_{\mathrm{pruned}}$. For any $\mathbf{m} \in M_{\mathrm{pruned}}$, let $\hat{f}_\mathbf{m}$ denote the corresponding pruned network. For any input $\mathbf{x} \in \mathcal{X} \subseteq \mathbb{R}^{d_0}$ with true class $t$, $\Delta(S, \mathbf{x}, t)$ denotes the worst-case margin deviation (WMD) such that*

$$\Delta(S, \mathbf{x}, t) \leq \max_{i \neq t} \|\mathbf{W}_{K+1}[t] - \mathbf{W}_{K+1}[i]\|_2 \cdot \max_{\mathbf{m} \in M_{\mathrm{pruned}}} \|\Delta\mathbf{x}_K(\mathbf{m})\|_2.$$

**Proof 1**
$$\Delta(S, \mathbf{x}, t) = \max_{\mathbf{m} \in M_{\text{pruned}}} \max_{i \neq t, i \in [C]} \left| (\hat{f}_{\mathbf{m}}^{(t)}(\mathbf{x}) - \hat{f}_{\mathbf{m}}^{(i)}(\mathbf{x})) - (f^{(t)}(\mathbf{x}) - f^{(i)}(\mathbf{x})) \right|$$

$$\overset{(i)}{=} \max_{\mathbf{m} \in M_{\text{pruned}}} \max_{i \neq t, i \in [C]} \left| (\mathbf{W}_{K+1}[t] - \mathbf{W}_{K+1}[i])^\top (\hat{\mathbf{x}}_K^p - \mathbf{x}_K) \right|$$

$$\overset{(ii)}{\leq} \max_{\mathbf{m} \in M_{\text{pruned}}} \max_{i \neq t, i \in [C]} \|\mathbf{W}_{K+1}[t] - \mathbf{W}_{K+1}[i]\|_2 \, \|\Delta \mathbf{x}_K(\mathbf{m})\|_2$$

$$\overset{(iii)}{=} \max_{i \neq t, i \in [C]} \|\mathbf{W}_{K+1}[t] - \mathbf{W}_{K+1}[i]\|_2 \cdot \max_{\mathbf{m} \in M_{\text{pruned}}} \|\Delta \mathbf{x}_K(\mathbf{m})\|_2 .$$

*where (i) follows from expanding the logits and using the fact that the output layer is not pruned. (ii) follows from the Cauchy-Schwarz inequality. (iii) holds since $\|\mathbf{W}_{K+1}[t] - \mathbf{W}_{K+1}[i]\|_2$ is independent of $\mathbf{m}$.*

The remaining challenge is to bound the layerwise output deviation $\|\Delta \mathbf{x}_k\|_2$ for $k \in [K]$. We consider three pruning scenarios—single-layer, selected multi-layer, and all-layer pruning—and derive tractable bounds inspired by Lipschitz-style perturbation analyses, but specialized to the discrete combinatorial space of structured neuron removal under layerwise sparsity budgets.

**Proposition 2 (Single-layer pruning bound)** *Let $f$ be a neural network with $K$ hidden fully connected layers and sparsity budget $S = \{s_1, \ldots, s_K\}$ such that only layer $\ell \in [K]$ is pruned, i.e., $s_\ell > 0$ and $s_k = 0$ for all $k \neq \ell$. Let $M_{\text{pruned}}$ denote the corresponding set of admissible pruning masks. For any input $\mathbf{x} \in \mathcal{X}$ with true label $t$, define*

$$\delta_\ell = \|tops_{s_\ell}(\mathbf{x}_\ell)\|_2,$$
$$\delta_k = \|\mathbf{W}_k\|_2 \, \delta_{k-1}, \qquad k = \ell + 1, \ldots, K.$$

*where $tops_{s_\ell}(\mathbf{x}_\ell)$ denotes the vector obtained by keeping the $s_\ell$ entries of $\mathbf{x}_\ell$ with largest absolute values and setting the rest to zero. The worst-case margin deviation (WMD) is bounded by*

$$\Delta(S, \mathbf{x}, t) \leq \max_{i \neq t, i \in [C]} \|\mathbf{W}_{K+1}[t] - \mathbf{W}_{K+1}[i]\|_2 \, \delta_K .$$

**Proof 2** *Since only layer $\ell$ is pruned, the hidden representations before layer $\ell$ are unchanged. Hence,*

$$\max_{\mathbf{m} \in M_{\text{pruned}}} \|\Delta \mathbf{x}_\ell\|_2 = \max_{\mathbf{m}_\ell \in M_\ell} \|\mathbf{m}_\ell \circ \mathbf{x}_\ell - \mathbf{x}_\ell\|_2 = \max_{\mathbf{m}_\ell \in M_\ell} \|\bar{\mathbf{m}}_\ell \circ \mathbf{x}_\ell\|_2 = \|tops_{s_\ell}(\mathbf{x}_\ell)\|_2 = \delta_\ell .$$

*For $k > \ell$, no further pruning is applied, so*

$$\max_{\mathbf{m} \in M_{\text{pruned}}} \|\Delta \mathbf{x}_k\|_2 = \max_{\mathbf{m} \in M_{pruned}} \|\sigma(\mathbf{W}_k \hat{\mathbf{x}}_{k-1}^p) - \sigma(\mathbf{W}_k \mathbf{x}_{k-1})\|_2$$

$$\overset{(i)}{\leq} \max_{\mathbf{m} \in M_{pruned}} \|\mathbf{W}_k (\hat{\mathbf{x}}_{k-1}^p - \mathbf{x}_{k-1})\|_2$$

$$\overset{(ii)}{\leq} \|\mathbf{W}_k\|_2 \max_{\mathbf{m} \in M_{pruned}} \|\Delta \mathbf{x}_{k-1}\|_2 .$$

*Thus $\max_{\mathbf{m} \in M_{pruned}} \|\Delta \mathbf{x}_k\|_2 \leq \delta_k, k = \ell + 1, \ldots, K$. The result then follows from Proposition 1. Here (i) follows from the $1-$Lipschitz property of $\sigma$. (ii) follows from the definition of the spectral norm.*

**Proposition 3 (All-layer pruning bound)** *Let $f$ be a neural network with $K$ hidden fully connected layers and sparsity budget $S = \{s_1, \ldots, s_K\}$, and let $M_{\text{pruned}}$ denote the corresponding set of admissible pruning masks. Assume that pruning is applied to all hidden layers, i.e., $s_k > 0$ for $k \in [K]$. For any input $\mathbf{x} \in \mathcal{X}$ with true label $t$, define*

$$\delta_1 = \|tops_{s_1}(\mathbf{x}_1)\|_2,$$
$$\delta_k = \|\mathbf{W}_k\|_2 \, \delta_{k-1} + \|tops_{s_k}(\mathbf{x}_k)\|_2, \qquad k = 2, \ldots, K,$$

*where $tops_{s_k}(\mathbf{x}_k)$ denotes the vector obtained by keeping the $s_k$ entries of $\mathbf{x}_k$ with largest absolute values and setting the rest to zero. Then the worst-case margin deviation (WMD) satisfies*

$$\Delta(S, \mathbf{x}, t) \leq \max_{i \neq t, i \in [C]} \|\mathbf{W}_{K+1}[t] - \mathbf{W}_{K+1}[i]\|_2 \, \delta_K .$$

**Proof 3** *The first-layer deviation satisfies*

$$\max_{\mathbf{m} \in M_{\text{pruned}}} \|\Delta \mathbf{x}_1\|_2 = \|tops_{s_1}(\mathbf{x}_1)\|_2 = \delta_1.$$

*For $k \geq 2$, by the triangle inequality, the masking equivalence under $\sigma(0) = 0$, the 1-Lipschitz property of $\sigma$, and the spectral norm bound,*

$$
\begin{aligned}
\max_{\mathbf{m} \in M_{\text{pruned}}} \|\Delta \mathbf{x}_k\|_2 &= \max_{\mathbf{m} \in M_{\text{pruned}}} \|\mathbf{m}_k \circ \sigma(\mathbf{W}_k \hat{\mathbf{x}}_{k-1}^p) - \sigma(\mathbf{W}_k \mathbf{x}_{k-1})\|_2 \\
&\leq \max_{\mathbf{m} \in M_{\text{pruned}}} \|\mathbf{m}_k \circ \sigma(\mathbf{W}_k \hat{\mathbf{x}}_{k-1}^p) - \mathbf{m}_k \circ \sigma(\mathbf{W}_k \mathbf{x}_{k-1})\|_2 + \max_{\mathbf{m}_k \in M_k} \|\mathbf{m}_k \circ \mathbf{x}_k - \mathbf{x}_k\|_2 \\
&\leq \max_{\mathbf{m} \in M_{\text{pruned}}} \|\sigma(\mathcal{P}_{\mathbf{m}_k}(\mathbf{W}_k)\hat{\mathbf{x}}_{k-1}^p) - \sigma(\mathcal{P}_{\mathbf{m}_k}(\mathbf{W}_k)\mathbf{x}_{k-1})\|_2 + \|tops_{s_k}(\mathbf{x}_k)\|_2 \\
&\leq \max_{\mathbf{m} \in M_{\text{pruned}}} \|\mathcal{P}_{\mathbf{m}_k}(\mathbf{W}_k)(\hat{\mathbf{x}}_{k-1}^p - \mathbf{x}_{k-1})\|_2 + \|tops_{s_k}(\mathbf{x}_k)\|_2 \\
&\leq \|\mathbf{W}_k\|_2 \max_{\mathbf{m} \in M_{\text{pruned}}} \|\Delta \mathbf{x}_{k-1}\|_2 + \|tops_{s_k}(\mathbf{x}_k)\|_2.
\end{aligned}
$$

*Applying Proposition 1 completes the proof. Details of the proof are provided in Appendix A.3*

The two terms in the upper bound above highlight that pruning affects the network through two mechanisms: a local perturbation introduced at the pruned layer, and its subsequent amplification through downstream layers. We now generalize to the case where pruning is applied to a subset of layers. In this case, if a layer is unpruned, the deviation follows the single-layer propagation rule; otherwise, it incorporates an additional local deviation term as in the all-layer pruning case.

**Proposition 4 (Selected multi-layer pruning bound)** *Let $f$ be a neural network with $K$ hidden fully connected layers and sparsity budget $S = \{s_1, \ldots, s_K\}$. Let $I_p \subseteq [K]$ denote the set of pruned layers, i.e., $s_k > 0$ for $k \in I_p$ and $s_k = 0$ otherwise, and let $\ell = \min I_p$ be the first pruned layer. Let $M_{\text{pruned}}$ denote the corresponding set of admissible masks. For any input $\mathbf{x} \in \mathcal{X}$ with true label $t$, define*

$$
\begin{aligned}
\delta_\ell &= \|tops_{s_\ell}(\mathbf{x}_\ell)\|_2, \\
\delta_k &= \begin{cases} \|\mathbf{W}_k\|_2 \, \delta_{k-1}, & k \notin I_p, \\ \|\mathbf{W}_k\|_2 \, \delta_{k-1} + \|tops_{s_k}(\mathbf{x}_k)\|_2, & k \in I_p, \end{cases} \quad k = \ell + 1, \ldots, K,
\end{aligned}
$$

*where $tops_{s_k}(\mathbf{x}_k)$ denotes the vector obtained by keeping the $s_k$ entries of $\mathbf{x}_k$ with largest absolute values and setting the rest to zero. The worst-case margin deviation (WMD) satisfies*

$$\Delta(S, \mathbf{x}, t) \leq \max_{i \neq t, i \in [C]} \|\mathbf{W}_{K+1}[t] - \mathbf{W}_{K+1}[i]\|_2 \, \delta_K.$$

Details of the proof are provided in Appendix A.4.

**Remark 4** *The proposed bounds can be computed efficiently via a single forward-style recursion once the required activations and spectral norms are available. The recursion itself scales linearly with depth. The spectral norm of each weight matrix can be estimated using $r$ steps of power iteration, the total cost is $O\left(\sum_{k=1}^K r \, d_k d_{k-1}\right)$. The additional cost of computing the local pruning terms $tops_{s_k}(\mathbf{x}_k)$ is at most $O(d_k \log d_k)$ per layer using sorting. This makes the bound substantially more efficient than MILP-based verification. The gain in scalability, however, comes at the expense of tightness because the use of spectral norms can be conservative, especially in deep networks where repeated multiplication may amplify over-approximation errors.*

## 4.3 Pruning-robust training loss

Building on the margin-bound analysis in Section 4.2, we design a training objective that directly promotes robustness against pruning by encouraging the condition that the worst-case margin deviation remains below

the nominal class margin. Recall that a sufficient condition for preserving the prediction under all admissible pruning masks is $\delta(S, \mathbf{x}, t) \leq \gamma(\mathbf{x}, t)$, where $\delta(S, \mathbf{x}, t)$ is the worst-case margin deviation bound and $\gamma(\mathbf{x}, t)$ is the margin between the true class logit and the largest competing logit.

Motivated by this condition, we introduce the following margin-aware loss:

$$\mathcal{L}_{\mathrm{pr}}\big(f(\mathbf{x}), y, \mathcal{F}_{\mathrm{pruned}}\big) = \mathrm{CE}\big(f(\mathbf{x}), y\big) + \lambda_1 \max\big(\delta(S, \mathbf{x}, t) - \gamma(\mathbf{x}, t), 0\big) + \lambda_2 \frac{\delta(S, \mathbf{x}, t)}{\max(\gamma(\mathbf{x}, t), \epsilon)}. \quad (10)$$

Here, $\mathrm{CE}(f(\mathbf{x}), y)$ is the standard cross-entropy loss, $\delta(S, \mathbf{x}, t)$ denotes the worst-case margin deviation bound, and $\gamma(\mathbf{x}, t)$ is the nominal margin. The hyperparameters $\lambda_1, \lambda_2 \geq 0$ control the trade-off between accuracy and pruning robustness.

The hinge term $\max(\delta - \gamma, 0)$ serves as the primary robustness term, as it directly penalizes violations of the sufficient condition $\delta \leq \gamma$ and therefore encourages margin preservation under pruning. The ratio term $\delta/\gamma_\epsilon$, with $\gamma_\epsilon = \max(\gamma, \epsilon)$, acts as an auxiliary scale-aware regularizer that normalizes the deviation relative to the nominal margin and remains invariant to global logit scaling.

**Remark 5** *The bound $\delta(S, \mathbf{x}, t)$ is composed of standard operations such as matrix norms, maximum, and top-k selection, all of which are piecewise differentiable. It can therefore be integrated directly into gradient-based optimization without introducing an additional differentiable approximation to the bound.*

*The training objective alone does not guarantee that the condition $\delta(S, \mathbf{x}, t) \leq \gamma(\mathbf{x}, t)$ holds for every sample after training. Certified robustness for a given input must be established through an independent post-hoc verification step, either by checking $\delta(S, \mathbf{x}, t) \leq \gamma(\mathbf{x}, t)$ or by applying the MILP verification procedure described in Section 4.1.*

## 5 Experiments

We evaluate the suitability of the proposed MILP verification framework (Section 4.1) and the pruning-robust training objective (Section 4.3) for Problems 1 and 2. First, we compare the verified accuracy under different pruning budgets for standard-trained models (trained with cross-entropy) and pruning-robust models (Section 5.1). We then analyze the verification tightness and scalability of MILP and margin-based bounds as the pruning budget increases (Section 5.2). Additionally, to assess the effect of pruning-robust training in adversarial settings, we evaluate models against three attacks: (i) projected gradient descent (PGD) adapted to pruning (Madry et al., 2017), (ii) $\ell_2$-based adversarial pruning, where neurons with the largest weight norms are removed, and (iii) random pruning (Section 5.3). Finally, we study the stability of model performance across different pruning strategies (Section 5.4).

All models are trained on MNIST (LeCun et al., 2010) and CIFAR-10 (Krizhevsky et al., 2009) using PyTorch. Experiments are conducted on a Dell Precision 7680 with an Intel i9-13950HX (32 Core) CPU, 64GB RAM, and NVIDIA GeForce RTX 4090 Laptop GPU with 16GB VRAM.[1]

### 5.1 Verification of pruning-robustness

We compute the verified accuracy of both robust-trained and standard-trained models on a fully connected network with four layers (128-64-32-10) using ReLU activations without bias terms. Verified accuracy is defined as the proportion of test samples for which the worst-case margin over all admissible pruning masks remains strictly positive. We report results obtained via MILP on 1000 MNIST test samples in Figure 1 (single-layer pruning), Figure 2 (multi-layer pruning), and Figure 3 (all-layer pruning). The MILP problems are solved to optimality, yielding exact certificates. Across all settings, robust training significantly improves certified robustness. For standard models, verified accuracy degrades rapidly as the sparsity budget increases, often collapsing at relatively small pruning levels. In contrast, robust-trained models maintain high verified accuracy over a substantially wider range of sparsity, particularly when pruning is applied to deeper layers.

---

[1]The source code will be made publicly available upon acceptance.

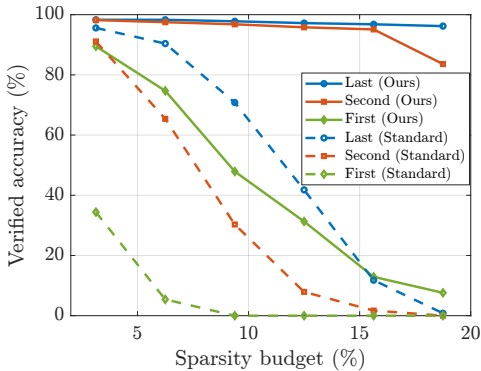

Figure 1: Verified accuracy on MNIST test set under single-layer pruning for robust-trained and standard models. "First", "Second", and "Last" denote pruning applied to the corresponding hidden layer.

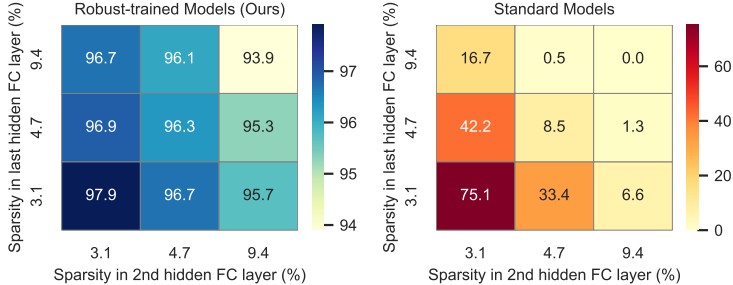

Figure 2: Verified accuracy on MNIST test set under multi-layer pruning settings for our trained-robust vs. standard models across different sparsity budgets in the last and second hidden FC layers.

We also observe that pruning the first hidden layer leads to a significantly larger degradation in verified accuracy compared to deeper layers. This is expected, as early layers directly encode input features, and pruning at this stage removes fundamental representations that cannot be recovered by subsequent layers.

Additional experiments are provided in Appendix A.5. In particular, more results for multi-layer pruning are shown in Figures 6 and 7. The trade-off between verified accuracy and clean accuracy, along with verification runtimes, is reported in Table 3. Computational overhead of robust training compared to standard training is discussed in Table 4. A grid search over $\lambda_1$ and $\lambda_2$ is presented in Figure 5, and additional results on CIFAR-10 are given in Figure 8.

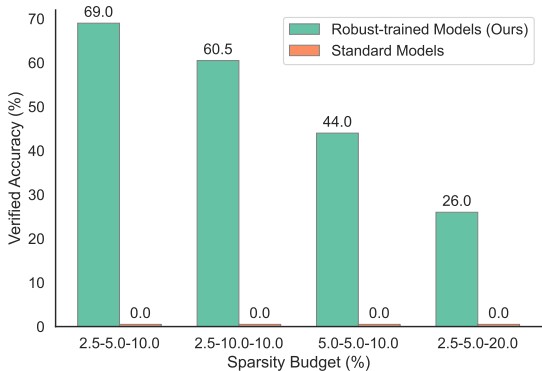

Figure 3: Verified accuracy on MNIST test set under all-layer pruning settings for trained-robust vs. standard models. Sparsity budgets denote pruning ratios to the first, second, last hidden FC layers, respectively.

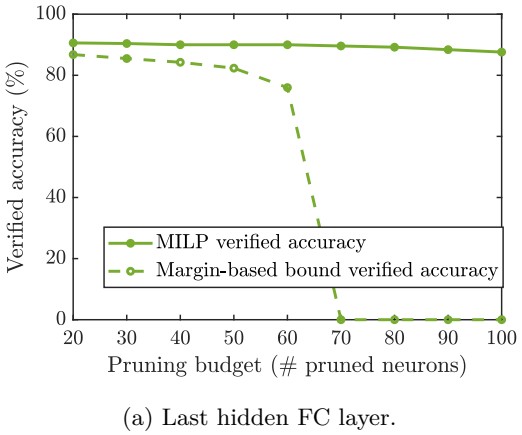

(a) Last hidden FC layer.

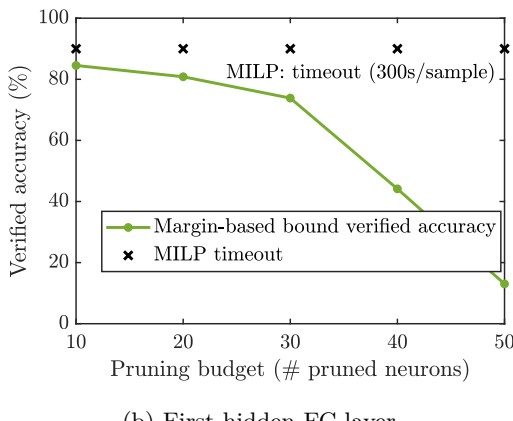

(b) First hidden FC layer.

Figure 4: Verified accuracy on CIFAR-10 (VGG11) comparing MILP and margin-based bounds. Left: pruning the last fully connected layer. Right: pruning the first fully connected layer—MILP times out beyond 10 pruned neurons (300s/sample), while the bound remains tractable and provides non-trivial certificates.

## 5.2 Verification Tightness and Scalability

We compare the verified accuracy of VGG11 on CIFAR-10 using MILP and margin-based bounds (Figure 4a and 4b). For pruning of the last fully connected layer (Figure 4a), margin-based bounds achieve comparable tightness to MILP under mild sparsity, but become increasingly loose as the pruning budget grows. Notably, in the tight-bound regime, margin-based bounds are significantly more efficient, verifying all 10k test images takes approximately 7 seconds on our machine, whereas MILP requires almost 5 hours (1.77s/image).

While MILP provides exact verification when solved to optimality, its practical scalability is limited. As shown in Figure 4b, the margin-based bound remains computationally efficient and yields non-trivial certificates, whereas MILP fails to terminate within a time limit of 300 seconds per sample. This is particularly pronounced in wider networks and when pruning is applied to early layers, where loose variable bounds and interactions with downstream activations increase the search complexity, rendering the problem intractable even for small pruning budgets. This highlights a trade-off between tightness and scalability and suggests a hybrid strategy in which analytical bounds are used to tighten intermediate variable bounds and pre-screen easy instances prior to selective exact MILP verification.

## 5.3 Robustness under Adversarial Pruning.

We observe that certified robustness is primarily achieved in the mild pruning regime (Section 5.1 and 5.2). As sparsity increases, the combinatorial nature of pruning induces increasingly adverse worst-case configurations, leading to a rapid collapse of certifiable margins even under exact MILP verification. To evaluate robustness beyond this regime, we consider adversarial pruning attacks, including PGD-based, $\ell_2$-based, and random pruning. We adapt PGD (Madry et al., 2017) by optimizing continuous neuron-importance scores and projecting them onto top-$k$ binary masks (with $k = d_k - s_k$), using a straight-through estimator for gradient propagation. The attack is run for 400 steps. For the $\ell_2$-based attack, we prune neurons with the largest row-wise weight norms, while for random pruning, we sample masks uniformly under the budget and report mean $\pm$ standard deviation over 1000 samples. We conduct this analysis on the SmallConv model on CIFAR-10, where pruning is applied to two hidden FC layers with 200 and 100 neurons. As these layers are relatively less redundant, pruning leads to more pronounced degradation.

Table 1 shows that our method consistently outperforms both standard training and SAM (Foret et al., 2020) across all pruning budgets and attack types. Here, SAM (Foret et al., 2020) is chosen as a flatness-based robustness baseline. The gains are especially significant under strong adversarial pruning (PGD), where our model maintains higher accuracy even at high sparsity levels. Improvements are also observed under random pruning, indicating that the learned robustness generalizes beyond worst-case attacks.

Table 1: Empirical robustness under L2-adversarial pruning, PGD attack, and Random pruning (mean $\pm$ std over 1000 samples). Pruned (fc1, fc2) denotes the number of neurons removed from the corresponding hidden FC layers.

| Pruned (fc1, fc2) | Model | Unpruned | L2-adv | PGD | Random $(\mu \pm \sigma)$ |
|---|---|---|---|---|---|
| 0-30 | Standard | 76.7 | 71.35 | 45.12 | 70.93 $\pm$ 1.42 |
| | SAM | 74.33 | 70.51 | 51.20 | 70.06 $\pm$ 0.92 |
| | **Ours** | 75.43 | **73.57** | **52.49** | **74.84 $\pm$ 0.88** |
| 0-40 | Standard | 76.7 | 67.36 | 24.69 | 67.98 $\pm$ 2.1 |
| | SAM | 74.33 | 67.21 | 35.97 | 68.52 $\pm$ 1.36 |
| | **Ours** | 75.33 | **73.8** | **46.10** | **74.56 $\pm$ 1.2** |
| 60-0 | Standard | 76.7 | 60.07 | 35.62 | 66.74 $\pm$ 1.5 |
| | SAM | 74.33 | 56.03 | 30.51 | 65.95 $\pm$ 1.27 |
| | **Ours** | 73.18 | **69.10** | **42.27** | **71.12 $\pm$ 0.81** |
| 80-0 | Standard | 76.7 | 52.8 | 19.04 | 62.28 $\pm$ 1.93 |
| | SAM | 74.33 | 47.96 | 20.30 | 62.15 $\pm$ 1.69 |
| | **Ours** | 73.48 | **65.29** | **33.62** | **69.9 $\pm$ 1.2** |
| 40-20 | Standard | 76.7 | 63.45 | 25.47 | 67.15 $\pm$ 1.71 |
| | SAM | 74.33 | 61.23 | 28.98 | 66.96 $\pm$ 1.27 |
| | **Ours** | 72.68 | **66.28** | **51.58** | **70.77 $\pm$ 0.86** |
| 60-30 | Standard | 76.7 | 54.85 | 10.45 | 61.16 $\pm$ 2.44 |
| | SAM | 74.33 | 51.61 | 12.23 | 62.09 $\pm$ 1.84 |
| | **Ours** | 72.88 | **60.81** | **48.42** | **69.57 $\pm$ 1.23** |

### 5.4 Analysis under Different Pruning Methods

We further evaluate robustness under common structured pruning heuristics, including L1-norm-based pruning (Li et al., 2017), L2-norm pruning (Han et al., 2015), Taylor (Molchanov et al., 2017b), and ActMean (Molchanov et al., 2017a). This experiment is conducted on the SmallConv model on CIFAR-10. As shown in Table 2, our method achieves the highest accuracy in most settings, with significantly smaller average accuracy drops compared to standard and SAM, especially under aggressive pruning. These results indicate that our training objective not only improves worst-case robustness under adversarial pruning, but also enhances stability under practical pruning heuristics.

We also evaluated Targeted Dropout (TD) (Gomez et al., 2019) as a pruning-aware baseline in Appendix A.7 and compared our methods with standard Dropout (Srivastava et al., 2014) in Appendix A.8. TD performs strongly under several benign pruning heuristics and exhibits relatively small clean-accuracy trade-offs. However, it is less reliable under adversarial pruning settings, where our method yields substantially stronger robustness. Standard Dropout similarly provides strong resilience to random pruning, while combining it with our robust training objective substantially improves robustness to adversarially optimized pruning masks and largely retains its resilience to random pruning.

## 6 Conclusion and Discussion

We proposed a verification framework based on MILP and margin-based bounds to certify robustness against all admissible pruning patterns under layerwise sparsity constraints, together with a training objective for pruning-robust models. While formal verification becomes challenging for larger networks, our results demonstrate non-trivial certified robustness in the mild pruning regime and strong empirical robustness under more aggressive pruning. Our framework focuses on neuron-level structured pruning in fully connected layers via row-masking operators. Extending the analysis to other structured pruning schemes, such as channel or filter pruning, is an interesting direction for future work. Finally, we note that our guarantees are deter-

Table 2: Empirical robustness under representative structured pruning heuristics: L2-norm, L1, Taylor, and ActMean pruning, along with the average accuracy drop from the corresponding unpruned model. Pruned (fc1, fc2) denotes the number of neurons removed from the corresponding hidden FC layers.

| Pruned (fc1, fc2) | Model | L2 | L1 | Taylor | ActMean | Avg. drop |
|---|---|---|---|---|---|---|
| 0-40 | Standard | 68.7 | 69.01 | 74.46 | 74.60 | 4.71 |
| | SAM | 66.95 | 66.90 | 72.08 | 70.29 | 5.28 |
| | **Ours** | **75.30** | **75.25** | **75.36** | **75.39** | **0.04** |
| 80-0 | Standard | 66.20 | 62.78 | 69.62 | 69.41 | 9.70 |
| | SAM | 67.54 | 66.70 | 70.36 | 69.92 | 5.70 |
| | **Ours** | **73.20** | **73.20** | **73.50** | **73.43** | **0.16** |
| 60-30 | Standard | 65.53 | 66.36 | 72.73 | 72.62 | 7.39 |
| | SAM | 65.54 | 65.15 | **72.89** | 72.23 | 5.38 |
| | **Ours** | **71.59** | **71.60** | 72.75 | 72.57 | **0.75** |
| 80-40 | Standard | 58.18 | 56.91 | 66.08 | 67.8 | 14.46 |
| | SAM | 59.73 | 57.97 | 66.72 | 64.84 | 12.02 |
| | **Ours** | **71.54** | **71.37** | **72.42** | **72.23** | **0.60** |

ministic and per-input; thus, the reported verified accuracy does not imply distribution-level generalization guarantees.

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

# A    Appendix

In this appendix section, we provide details of Proposition 1, 2, 3, 4. Additionally, we present details of the experiments and additional experiments mentioned in the main paper.

## A.1    Proof of Proposition 1

From the definition of the worst-case pairwise class margin in the equation 6, we have:

$$
\begin{aligned}
\Delta(S, \mathbf{x}, t) &= \max_{\mathbf{m} \in M_{\text{pruned}}} \max_{i \neq t, i \in [C]} \left| (\hat{f}_{\mathbf{m}}^{(t)}(\mathbf{x}) - \hat{f}_{\mathbf{m}}^{(i)}(\mathbf{x})) - (f^{(t)}(\mathbf{x}) - f^{(i)}(\mathbf{x})) \right| \\
&\overset{(i)}{=} \max_{\mathbf{m} \in M_{\text{pruned}}} \max_{i \neq t, i \in [C]} \left| \left( \mathbf{W}_{K+1}[t]^{\top} \cdot \hat{\mathbf{x}}_K^p - \mathbf{W}_{K+1}[i]^{\top} \cdot \hat{\mathbf{x}}_K^p \right) - \left( \mathbf{W}_{K+1}[t]^{\top} \cdot \mathbf{x}_K - \mathbf{W}_{K+1}[i]^{\top} \cdot \mathbf{x}_K \right) \right| \\
&= \max_{\mathbf{m} \in M_{\text{pruned}}} \max_{i \neq t, i \in [C]} \left| (\mathbf{W}_{K+1}[t] - \mathbf{W}_{K+1}[i])^{\top} (\hat{\mathbf{x}}_K^p - \mathbf{x}_K) \right| \\
&\overset{(ii)}{\leq} \max_{\mathbf{m} \in M_{\text{pruned}}} \max_{i \neq t, i \in [C]} \| \mathbf{W}_{K+1}[t] - \mathbf{W}_{K+1}[i] \|_2 \| \Delta \mathbf{x}_K \|_2 \\
&\overset{(iii)}{=} \max_{i \neq t, i \in [C]} \| \mathbf{W}_{K+1}[t] - \mathbf{W}_{K+1}[i] \|_2 \cdot \max_{\mathbf{m} \in M_{\text{pruned}}} \| \Delta \mathbf{x}_K \|_2 .
\end{aligned}
\tag{11}
$$

In the proof above, equality (i) is because there is no pruning in the output layer. In (ii), we apply the Cauchy-Schwarz inequality: $|\mathbf{a}^{\top} \mathbf{b}| \leq \|\mathbf{a}\|_2 \|\mathbf{b}\|_2$, where $\mathbf{a} = \mathbf{W}_{K+1}[t] - \mathbf{W}_{K+1}[i]$ and $\mathbf{b} = \hat{\mathbf{x}}_K^p - \mathbf{x}_K$. and (iii) is because $\|\mathbf{W}_{K+1}[t] - \mathbf{W}_{K+1}[i]\|_2$ does not depend on the mask $\mathbf{m}$, it can be taken outside the maximization.

Since $\max_{i \neq t, i \in [C]} \|\mathbf{W}_{K+1}[t] - \mathbf{W}_{K+1}[i]\|_2$ can be computed as the maximum pairwise Euclidean distance between row $t$ and all other rows of $\mathbf{W}_{K+1}$, the remaining challenge in deriving an upper bound on the WMD is to bound the worst-case layer-wise output deviation norm $\|\Delta \mathbf{x}_K\|_2$. The corresponding results are presented in Propositions 2, 3, and 4, with detailed proofs provided below.

## A.2    Proof of Single-layer Pruning Bound (For Proposition 2)

Let $\ell$ be the only pruned hidden layer with sparsity budget $s_\ell$ ( $s_k = 0$ for all $k \neq \ell$). Since only layer $\ell$ is pruned, the representations before layer $\ell$ remain unchanged, i.e., $\hat{\mathbf{x}}_{\ell-1}^p = \mathbf{x}_{\ell-1}$. Hence,

$$
\begin{aligned}
\max_{\mathbf{m} \in M_{\text{pruned}}} \| \Delta \mathbf{x}_\ell \|_2 &= \max_{\mathbf{m}_\ell \in M_\ell} \| \mathbf{m}_\ell \circ \mathbf{x}_\ell - \mathbf{x}_\ell \|_2 \\
&= \max_{\mathbf{m}_\ell \in M_\ell} \| (\mathbf{m}_\ell - \mathbf{1}) \circ \mathbf{x}_\ell \|_2 \\
&= \max_{\mathbf{m}_\ell \in M_\ell} \| \bar{\mathbf{m}}_\ell \circ \mathbf{x}_\ell \|_2 \\
&= \| tops_{s_\ell}(\mathbf{x}_\ell) \|_2 = \delta_\ell ,
\end{aligned}
$$

where $\bar{\mathbf{m}}_\ell = \mathbf{1} - \mathbf{m}_\ell$, and the maximum is attained by selecting the $s_\ell$ entries of largest magnitude.

For $k > \ell$, no further pruning is applied, so

$$
\begin{aligned}
\max_{\mathbf{m} \in M_{\text{pruned}}} \| \Delta \mathbf{x}_k \|_2 &= \max_{\mathbf{m} \in M_{\text{pruned}}} \| \sigma(\mathbf{W}_k \hat{\mathbf{x}}_{k-1}^p) - \sigma(\mathbf{W}_k \mathbf{x}_{k-1}) \|_2 \\
&\overset{(i)}{\leq} \max_{\mathbf{m} \in M_{\text{pruned}}} \| \mathbf{W}_k (\hat{\mathbf{x}}_{k-1}^p - \mathbf{x}_{k-1}) \|_2 \\
&\overset{(ii)}{\leq} \| \mathbf{W}_k \|_2 \max_{\mathbf{m} \in M_{\text{pruned}}} \| \Delta \mathbf{x}_{k-1} \|_2 ,
\end{aligned}
\tag{12}
$$

where (i) uses the 1-Lipschitz property of $\sigma$, and (ii) follows from the definition of the spectral norm.

By recursion, $\max_{\mathbf{m} \in M_{\text{pruned}}} \| \Delta \mathbf{x}_k \|_2 \leq \delta_k$ for $k = \ell + 1, \dots, K$. The result then follows from Proposition 1.

### A.3 Proof for All-layer Pruning Bound - Proposition 3

For the first hidden layer,

$$\max_{\mathbf{m} \in M_{\text{pruned}}} \|\Delta \mathbf{x}_1\|_2 = \max_{\mathbf{m}_1 \in M_1} \|\mathbf{m}_1 \circ \mathbf{x}_1 - \mathbf{x}_1\|_2 = \|tops_{s_1}(\mathbf{x}_1)\|_2 = \delta_1.$$

For each layer $k \geq 2$,

$$
\begin{aligned}
\max_{\mathbf{m} \in M_{\text{pruned}}} \|\Delta \mathbf{x}_k\|_2 &= \max_{\mathbf{m} \in M_{\text{pruned}}} \|\mathbf{m}_k \circ \sigma(\mathbf{W}_k \hat{\mathbf{x}}_{k-1}^p) - \sigma(\mathbf{W}_k \mathbf{x}_{k-1})\|_2 \\
&\overset{(i)}{\leq} \max_{\mathbf{m} \in M_{\text{pruned}}} \|\mathbf{m}_k \circ \sigma(\mathbf{W}_k \hat{\mathbf{x}}_{k-1}^p) - \mathbf{m}_k \circ \sigma(\mathbf{W}_k \mathbf{x}_{k-1})\|_2 + \max_{\mathbf{m}_k \in M_k} \|\mathbf{m}_k \circ \mathbf{x}_k - \mathbf{x}_k\|_2 \\
&\overset{(ii)}{\leq} \max_{\mathbf{m} \in M_{\text{pruned}}} \|\sigma(\mathcal{P}_{\mathbf{m}_k}(\mathbf{W}_k) \hat{\mathbf{x}}_{k-1}^p) - \sigma(\mathcal{P}_{\mathbf{m}_k}(\mathbf{W}_k) \mathbf{x}_{k-1})\|_2 + \|tops_{s_k}(\mathbf{x}_k)\|_2 \\
&\overset{(iii)}{\leq} \max_{\mathbf{m} \in M_{\text{pruned}}} \|\mathcal{P}_{\mathbf{m}_k}(\mathbf{W}_k)(\hat{\mathbf{x}}_{k-1}^p - \mathbf{x}_{k-1})\|_2 + \|tops_{s_k}(\mathbf{x}_k)\|_2 \\
&\overset{(iv)}{\leq} \|\mathbf{W}_k\|_2 \max_{\mathbf{m} \in M_{\text{pruned}}} \|\Delta \mathbf{x}_{k-1}\|_2 + \|tops_{s_k}(\mathbf{x}_k)\|_2.
\end{aligned}
\tag{13}
$$

Here, (i) follows from the triangle inequality after adding and subtracting $\mathbf{m}_k \circ \sigma(\mathbf{W}_k \mathbf{x}_{k-1})$. Step (ii) uses the equivalence between activation masking and row masking under $\sigma(0) = 0$, together with the definition of $tops_{s_k}(\mathbf{x}_k)$. Step (iii) follows from the 1-Lipschitz property of $\sigma$. Step (iv) follows from the definition of the spectral norm and the inequality $\|\mathcal{P}_{\mathbf{m}_k}(\mathbf{W}_k)\|_2 \leq \|\mathbf{W}_k\|_2$. Thus, by recursion,

$$\max_{\mathbf{m} \in M_{\text{pruned}}} \|\Delta \mathbf{x}_k\|_2 \leq \delta_k, \qquad k = 1, \dots, K.$$

Applying Proposition 1 yields the desired result.

### A.4 Proof for Multi-layer Pruning Bound - Proposition 4

Let $I_p \subseteq [K]$ denote the set of pruned hidden layers, and let $\ell = \min I_p$ be the first pruned layer. Since no pruning is applied before layer $\ell$, the hidden representations up to layer $\ell-1$ are unchanged, i.e., $\hat{\mathbf{x}}_{\ell-1}^p = \mathbf{x}_{\ell-1}$. Hence,

$$
\begin{aligned}
\max_{\mathbf{m} \in M_{\text{pruned}}} \|\Delta \mathbf{x}_\ell\|_2 &= \max_{\mathbf{m}_\ell \in M_\ell} \|\mathbf{m}_\ell \circ \mathbf{x}_\ell - \mathbf{x}_\ell\|_2 \\
&= \max_{\mathbf{m}_\ell \in M_\ell} \|\bar{\mathbf{m}}_\ell \circ \mathbf{x}_\ell\|_2 \\
&= \|tops_{s_\ell}(\mathbf{x}_\ell)\|_2 = \delta_\ell,
\end{aligned}
$$

where $\bar{\mathbf{m}}_\ell = \mathbf{1} - \mathbf{m}_\ell$.

For any layer $k > \ell$, two cases arise.

**Case 1:** $k \notin I_p$. No pruning is applied at layer $k$, we apply equation 12 from the single-layer pruning bound, that is,

$$
\begin{aligned}
\max_{\mathbf{m} \in M_{\text{pruned}}} \|\Delta \mathbf{x}_k\|_2 &= \max_{\mathbf{m} \in M_{\text{pruned}}} \|\sigma(\mathbf{W}_k \hat{\mathbf{x}}_{k-1}^p) - \sigma(\mathbf{W}_k \mathbf{x}_{k-1})\|_2 \\
&\leq \|\mathbf{W}_k\|_2 \max_{\mathbf{m} \in M_{\text{pruned}}} \|\Delta \mathbf{x}_{k-1}\|_2.
\end{aligned}
$$

Thus, $\max_{\mathbf{m} \in M_{\text{pruned}}} \|\Delta \mathbf{x}_k\|_2 \leq \|\mathbf{W}_k\|_2 \delta_{k-1} = \delta_k$.

**Case 2:** $k \in I_p$. Pruning is applied at layer $k$, we apply equation 13 from the all-layer pruning bound

$$\max_{\mathbf{m} \in M_{\text{pruned}}} \|\Delta \mathbf{x}_k\|_2 = \max_{\mathbf{m} \in M_{\text{pruned}}} \|\mathbf{m}_k \circ \sigma(\mathbf{W}_k \hat{\mathbf{x}}_{k-1}^p) - \sigma(\mathbf{W}_k \mathbf{x}_{k-1})\|_2$$

$$\leq \|\mathbf{W}_k\|_2 \max_{\mathbf{m} \in M_{\text{pruned}}} \|\Delta \mathbf{x}_{k-1}\|_2 + \|tops_{s_k}(\mathbf{x}_k)\|_2.$$

Therefore,

$$\max_{\mathbf{m} \in M_{\text{pruned}}} \|\Delta \mathbf{x}_k\|_2 \leq \|\mathbf{W}_k\|_2 \delta_{k-1} + \|tops_{s_k}(\mathbf{x}_k)\|_2 = \delta_k.$$

Combining the two cases, we obtain recursively that

$$\max_{\mathbf{m} \in M_{\text{pruned}}} \|\Delta \mathbf{x}_k\|_2 \leq \delta_k, \qquad k = \ell, \dots, K.$$

The result then follows from Proposition 1.

### A.5 Additional Experiments on MNIST dataset

For experiments on MNIST (LeCun et al., 2010) on the architecture of three hidden layers 128-64-32-10 reported in the manuscript, image pixel values were rescaled from the range $[0, 255]$ to $[0, 1]$ without applying any additional preprocessing. For standard training, the neural network was trained using Adam algorithm (Adam et al., 2014) with a weight decay of $1 \times 10^{-3}$. The model was trained for 120 epochs with a batch size of 128 on an NVIDIA GeForce RTX 4090 Laptop GPU.

For robust training, we also used the Adam algorithm with the same hyperparameters as defined in the standard training process. We leverage a learning curriculum by scheduling the values of $\lambda_1$ and $\lambda_2$. Specifically, after 5 warm-up epochs, we linearly increase the scaling factor $\lambda_1, \lambda_2$ from 0 to the target values of $\lambda_1$ and $\lambda_2$ over next 10 epochs. The schedule is applied at every batch update, resulting in a total of $10\times$ (number of batches per epoch) linear steps. The target values of $\lambda_1$ and $\lambda_2$ are selected from a grid search (Figure 5). Moreover, we also propose a more principled way to calibrate $\lambda_1$ and $\lambda_2$ in Section A.6.

We evaluate the verified accuracy on a subset of the MNIST test set using our MILP verification method. For each test sample, the optimization problem is solved using Gurobi 12.0.3 with a 300-second time limit. If Gurobi fails to find a solution within the allotted time or the optimization is unsuccessful due to an out-of-memory error, the sample is classified as non-robust.

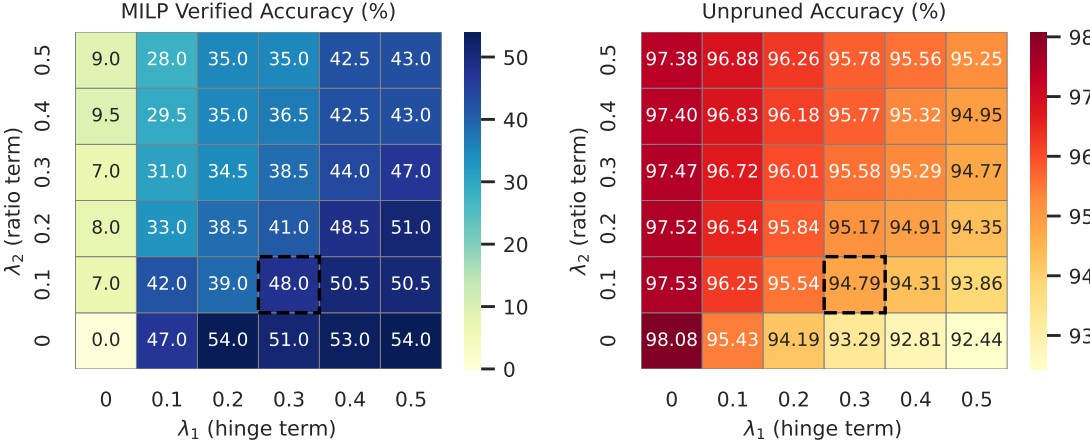

Figure 5: Grid search over $\lambda_1$ (hinge term) and $\lambda_2$ (ratio term) under 10% neuron pruning at the first hidden FC layer. The left heatmap shows MILP verified accuracy (%), and the right heatmap shows clean accuracy (%). The selected configuration $(\lambda_1, \lambda_2) = (0.3, 0.1)$ achieves a favorable balance between verified and unpruned performance.

Table 3: Representative clean–verified accuracy trade-offs under different pruning settings for the 128–64–32–10 network on MNIST. Sparsity $(s_1, s_2, s_3)$ denotes the percentage of neurons pruned in the first, second, and last hidden layers, respectively. The corresponding number of pruned neurons is obtained by rounding to the nearest integer.

| Sparsity ratio (%) | Clean acc. (%) | | Verified acc. (%) | | Runtime (s/sample) | |
|---|---|---|---|---|---|---|
| | Robust | Standard | Robust | Standard | Robust | Standard |
| $(0, 0, 10)$ | 98.19 | 98.08 | **96.2** | 70.8 | **0.14** | 0.16 |
| $(0, 0, 20)$ | 98.13 | 98.08 | **96.2** | 0.8 | 0.56 | **0.16** |
| $(0, 10, 0)$ | 97.43 | 98.08 | **96.8** | 30.3 | 0.55 | **0.24** |
| $(0, 20, 0)$ | 97.64 | 98.08 | **83.6** | 0.0 | 0.7 | **0.46** |
| $(10, 0, 0)$ | 94.79 | 98.08 | **47.9** | 0.0 | **4.85** | 17.89 |
| $(13, 0, 0)$ | 94.99 | 98.08 | **31.3** | 0.0 | **4.22** | 24.41 |
| $(0, 10, 10)$ | 97.38 | 98.08 | **93.9** | 0.0 | 0.99 | **0.50** |
| $(6, 0, 10)$ | 95.22 | 98.08 | **56.0** | 0.0 | **18.74** | 30.29 |
| $(6, 10, 0)$ | 95.14 | 98.08 | **43.90** | 0.0 | 38.66 | **18.18** |
| $(5, 5, 10)$ | 94.89 | 98.08 | **44.0** | 0.0 | 70.44 | **22.34** |

**Grid Search over $\lambda_1$ and $\lambda_2$**  To select $\lambda_1$ and $\lambda_2$ for the MNIST experiments reported in the main paper, we perform a grid search over the range $[0, 0.5]$ under a representative single-layer pruning setting (10% pruning at the first hidden layer). The results are shown in Figure 5, where verified accuracy is evaluated on 200 MNIST test samples using MILP.

The row $\lambda_2 = 0$ isolates the effect of the hinge term alone, while the column $\lambda_1 = 0$ isolates the ratio term, effectively serving as an ablation study over the two loss components. This comparison reveals that verified robustness is primarily driven by the hinge term $\lambda_1$: increasing $\lambda_1$ leads to substantial improvements in verified accuracy, whereas the ratio term $\lambda_2$ has a comparatively smaller but non-negligible effect, mainly refining performance for a fixed $\lambda_1$. This suggests that enforcing an absolute margin constraint (via the hinge term) is more critical for pruning robustness than relative normalization.

We also observe a clear trade-off between verified and clean accuracy: larger values of $\lambda_1$ improve robustness but reduce clean accuracy. The ratio term $\lambda_2$ provides additional flexibility in navigating this trade-off without substantially degrading verified accuracy. Based on these observations, we select $(\lambda_1, \lambda_2) = (0.3, 0.1)$ as a configuration that achieves a favorable balance between verified robustness and clean accuracy. All MNIST results reported in the main paper use this configuration; setting-specific tuning could further improve results for individual pruning budgets.

**Robustness-Accuracy trade-off and Verification Runtime Analysis**  Table 3 highlights a clear trade-off between clean (unpruned) and verified accuracy across different pruning settings. Across all settings, the robust-trained model consistently achieves substantially higher verified accuracy than the standard model, particularly under aggressive and multi-layer pruning, while incurring only a modest reduction in clean accuracy (within 4%).

We also report the average verification runtime per MNIST sample. In many cases where the robust model attains high verified accuracy, its verification runtime is higher than that of the standard model, whose verified accuracy often collapses to zero. This suggests that finding counterexamples is easier for standard models, allowing the verifier to terminate earlier. In contrast, when the robust model achieves moderate verified accuracy, its runtime is often lower than that of the standard model. One possible explanation is that robust training leads to tighter intermediate bounds, which reduces the effective search space and improves verification efficiency.

**Training Runtime Analysis**  Table 4 shows that the computational overhead of robust training depends on the model architecture. Robust training introduces additional computations, including spectral norm

Table 4: Per-epoch training run time (in seconds) averaged over 20 epochs for the standard and robust models across different datasets and architectures.

|  | Standard model | Robust model |
|---|---|---|
| MLP (MNIST) | 1.012 sec. | 2.645 sec. |
| SmallConv (CIFAR-10) | 5.656 sec. | 5.717 sec. |

estimation with complexity $\mathcal{O}(kd^2)$ for fully connected layers, where $k$ is the number of power iterations and $d$ is the layer width. However, in convolutional neural networks (CNNs), the overall training cost is dominated by convolutional operations, which typically scale as $\mathcal{O}(HWC_{\text{in}}C_{\text{out}}K^2)$. As a result, the additional overhead from the robustness terms becomes negligible in CNN training. In contrast, for smaller architectures such as MLPs, where the baseline computational cost is low, the relative overhead becomes more noticeable.

**More experiments on multi-layer pruning** More experiments on multi-layer pruning are shown in Figure 6 and 7, showing that robust training significantly improves certified robustness. For standard models, verified accuracy degrades rapidly as the sparsity budget increases, often collapsing at relatively small pruning budgets.

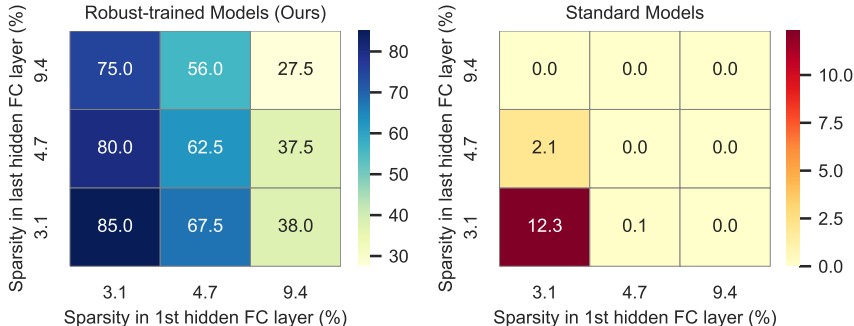

Figure 6: Verified accuracy on MNIST test set under multi-layer pruning settings for our trained-robust vs. standard models across different sparsity budgets in the first and last hidden FC layers.

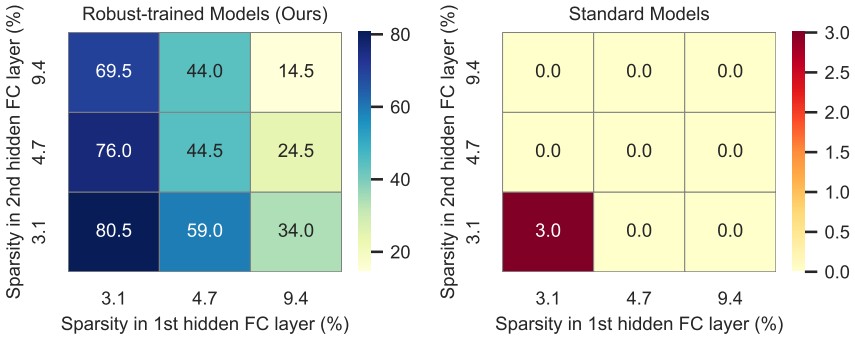

Figure 7: Verified accuracy on MNIST test set under multi-layer pruning settings for our trained-robust vs. standard models across different sparsity budgets in the first and second hidden FC layers.

## A.6 Calibration of $\lambda_1$ and $\lambda_2$ in SmallConv-CIFAR-10

For CIFAR-10 (Krizhevsky et al., 2009), we use a SmallConv architecture consisting of three convolutional layers (with 8, 16, and 32 filters) followed by two hidden fully connected (FC) layers of sizes 200 and 100. Input images are normalized using mean $[0.4914, 0.4822, 0.4465]$ and standard deviation $[0.2470, 0.2435, 0.2616]$. Structured pruning is applied only to the FC layers.

To stabilize optimization, we adopt a curriculum schedule for $\lambda_1$ and $\lambda_2$: after 20 warm-up epochs, both coefficients are linearly increased from 0 to their target values over the next 20 epochs, with updates applied at each batch.

We adopt a consistent tuning strategy across datasets, where grid search is used to select $\lambda_1$ and $\lambda_2$. For CIFAR-10, we further leverage gradient scale analysis to guide the search space, providing a more principled and scalable hyperparameter calibration. Specifically, we estimate the relative magnitudes and alignment of the gradients of each loss component. For pruning on the last FC layer (0–10 setting), we observe that $g_1/g_{\mathrm{CE}} \approx 0.15$ and $g_2/g_{\mathrm{CE}} \approx 0.0025$, with $\cos(\mathrm{CE}, L_1) \approx -0.06$ and $\cos(\mathrm{CE}, L_2) \approx 0.29$. This suggests that the hinge term $L_1$ can be increased moderately without significantly degrading clean accuracy.

We therefore search $\lambda_1$ and $\lambda_2$ around these ratios. Figure 8 shows the grid search results. We observe trends consistent with MNIST: (i) verified robustness is primarily driven by $\lambda_1$, (ii) increasing $\lambda_1$ improves robustness at a mild cost to clean accuracy, and (iii) varying $\lambda_2$ provides additional flexibility in the verified–clean trade-off. Based on these results, we select $(\lambda_1, \lambda_2) = (10^{-2}, 0.3)$ for pruning on the last fully connected layer. A similar strategy is applied for the first fully connected layer and all-layer pruning experiments, where we select $(\lambda_1, \lambda_2) = (10^{-2}, 0.05)$.

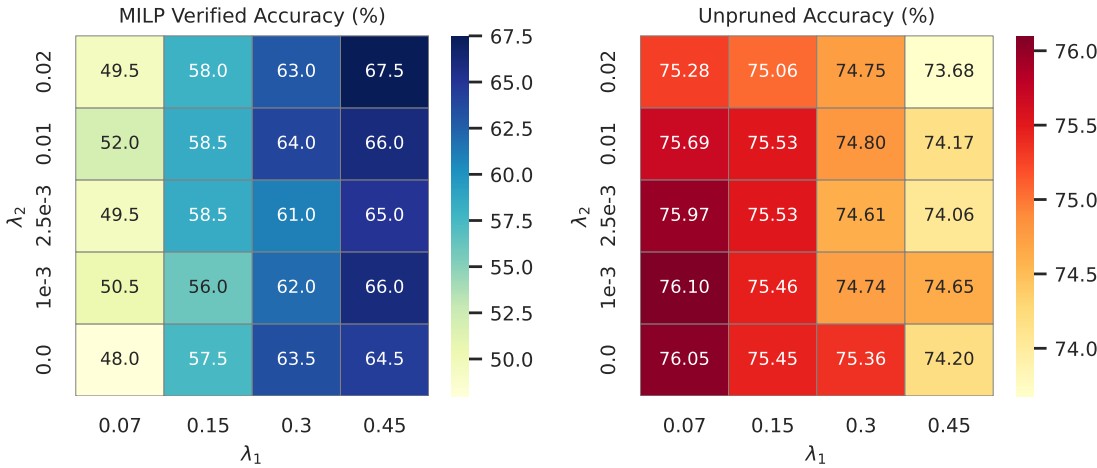

Figure 8: Grid search over $\lambda_1$ (hinge term) and $\lambda_2$ (ratio term) under 10% neuron pruning at the last hidden FC layer. The left heatmap shows CIFAR-10 verified accuracy (%), and the right heatmap shows clean accuracy (%)

A trade-off between clean (unpruned) and verified accuracy across different pruning settings on Smallconv-CIFAR-10 is shown in Table 5.

## A.7 Targeted-dropout Model Performance under Different Pruning Methods

We evaluate Targeted Dropout (TD) (Gomez et al., 2019) as a pruning-aware baseline that encourages resilience to likely removable units during training. Table 6 shows a clear separation between heuristic (benign) pruning and worst-case pruning robustness. While TD performs well under non-adversarial pruning criteria, it is less reliable under adversarial pruning, where our method achieves substantially stronger robustness. Notably, this robustness gap widens as the pruning budget increases, highlighting the vulnerability of TD under more aggressive pruning.

Table 5: Verified accuracy under different settings for SmallConv on 1000 CIFAR-10 test images. Sparsity $(s_1, s_2)$ denotes the number of neurons pruned in the first and last hidden FC layers, respectively.

| Sparsity | Clean acc. (%) | | Verified acc. (%) | |
|---|---|---|---|---|
| | Robust | Standard | Robust | Standard |
| (0, 5) | 74.99 | 76.70 | **69.0** | 19.9 |
| (0, 10) | 74.80 | 76.70 | **63.5** | 0.7 |
| (0, 15) | 75.84 | 76.70 | **18.0** | 0.0 |
| (5, 0) | 70.66 | 76.70 | **65.6** | 2.5 |
| (10, 0) | 69.79 | 76.70 | **45.0** | 0.1 |
| (15, 0) | 68.25 | 76.70 | **33.4** | 0.0 |
| (5, 10) | 70.32 | 76.70 | **32.5** | 0.0 |

Table 6: Performance under L1-norm, L2-norm, Taylor, ActMean, PGD attack, and random pruning (mean ± std over 1000 samples).

| Pruned (fc1, fc2) | Model | Unpruned | L1 | L2 | Taylor | ActMean | PGD | Random ($\mu \pm \sigma$) |
|---|---|---|---|---|---|---|---|---|
| 0-30 | Standard | 76.7 | 72.17 | 72.0 | 75.88 | 75.64 | 45.12 | 70.93 ± 1.42 |
| | TD | 75.79 | 75.12 | 75.10 | 75.65 | 75.52 | 35.50 | 67.09 ± 3.09 |
| | **Ours** | 75.43 | 75.48 | 75.48 | 75.42 | 75.42 | **52.49** | **74.84 ± 0.88** |
| 0-40 | Standard | 76.7 | 69.01 | 68.7 | 74.46 | 74.60 | 24.69 | 67.98 ± 2.1 |
| | TD | 76.87 | 76.15 | 76.23 | 76.63 | 76.18 | 28.82 | 61.25 ± 4.96 |
| | **Ours** | 75.33 | 75.25 | 75.30 | 75.36 | 75.39 | **46.10** | **74.56 ± 1.2** |
| 60-0 | Standard | 76.7 | 69.91 | 70.51 | 73.57 | 73.38 | 35.62 | 66.74 ± 1.5 |
| | TD | 76.81 | 76.65 | 76.65 | 76.73 | 76.56 | 30.92 | 66.52 ± 1.74 |
| | **Ours** | 73.18 | 72.98 | 72.94 | 73.18 | 73.18 | **42.27** | **71.12 ± 0.81** |
| 80-0 | Standard | 76.7 | 62.78 | 66.2 | 69.62 | 69.41 | 19.04 | 62.28 ± 1.93 |
| | TD | 76.36 | 75.90 | 75.99 | 76.0 | 75.69 | 12.52 | 61.79 ± 2.25 |
| | **Ours** | 73.48 | 73.20 | 73.20 | 73.50 | 73.43 | **33.62** | **69.9 ± 1.2** |
| 60-30 | Standard | 76.7 | 66.36 | 66.53 | 72.73 | 72.62 | 10.45 | 61.16 ± 2.44 |
| | TD | 76.45 | 75.85 | 75.71 | 76.04 | 75.76 | 10.23 | 60.36 ± 2.78 |
| | **Ours** | 72.88 | 71.37 | 71.54 | 72.42 | 72.23 | **48.42** | **69.57 ± 1.23** |

TD also exhibits higher variance under random pruning, suggesting limited generalization beyond the specific pruning heuristic used during training. We attribute this gap to the training objectives: TD promotes robustness to likely removable units, whereas our method directly optimizes a bound on the worst-case pairwise margin degradation over all pruning masks within a given sparsity budget, leading to stronger and more generalizable robustness.

## A.8 Comparison and Combination with Standard Dropout

We evaluate three training strategies: (i) standard dropout, which applies stochastic neuron masking during training and serves as a natural baseline for robustness to random pruning (Dropout); (ii) our margin-based robust training objective (Ours); and (iii) their combination (Ours+Dropout).

Table 7 shows that dropout and our robust objective exhibit complementary behavior. Dropout achieves the highest random-pruning accuracy for most budgets, generally with low variance, which is consistent with its use of stochastic neuron masking during training. Under PGD-based pruning, however, Ours outperforms Dropout across all evaluated multi-layer pruning budgets.

Table 7: Comparison of standard dropout, our pruning-robust training objective, and their combination under PGD-based and random pruning. Worst PGD reports the lowest accuracy obtained over the evaluated attack configurations. Random-pruning accuracy is reported as mean ± standard deviation over 1,000 sampled masks. Pruned (fc1, fc2) denotes the number of neurons removed from the corresponding hidden FC layers. All values are accuracies (%).

| Pruned (fc1, fc2) | Model | Unpruned | Worst PGD | Random ($\mu \pm \sigma$) |
|---|---|---|---|---|
| 0-30 | Dropout | 76.23 | **71.15** | **76.49 ± 0.25** |
| | **Ours** | 75.43 | 52.49 | 75.14 ± 0.81 |
| | **Ours+Dropout** | **77.36** | 60.36 | 76.01 ± 0.16 |
| 0-40 | Dropout | 74.99 | 67.46 | 75.94 ± 0.25 |
| | **Ours** | 75.33 | 46.10 | 74.73 ± 1.56 |
| | **Ours+Dropout** | **76.99** | **67.64** | **76.72 ± 0.37** |
| 60-0 | Dropout | **76.76** | 55.00 | **74.29 ± 0.42** |
| | **Ours** | 73.18 | 42.27 | 71.55 ± 0.84 |
| | **Ours+Dropout** | 72.90 | **56.23** | 72.13 ± 0.33 |
| 80-0 | Dropout | **74.99** | 46.14 | **72.31 ± 0.47** |
| | **Ours** | 73.48 | 33.62 | 70.47 ± 1.19 |
| | **Ours+Dropout** | 71.65 | **53.58** | 70.75 ± 0.37 |
| 40-20 | Dropout | **76.66** | 49.77 | **74.30 ± 0.42** |
| | **Ours** | 72.68 | 51.58 | 70.91 ± 0.75 |
| | **Ours+Dropout** | 72.87 | **65.63** | 72.06 ± 0.33 |
| 50-25 | Dropout | **75.94** | 48.12 | **73.30 ± 0.48** |
| | **Ours** | 72.80 | 48.90 | 70.50 ± 1.06 |
| | **Ours+Dropout** | 72.99 | **62.56** | 71.83 ± 0.42 |
| 60-30 | Dropout | **74.99** | 42.96 | **72.36 ± 0.50** |
| | **Ours** | 72.88 | 48.42 | 69.57 ± 1.23 |
| | **Ours+Dropout** | 72.22 | **56.51** | 71.27 ± 0.50 |

Combining the two objectives provides the strongest empirical worst-case robustness overall, particularly under multi-layer pruning. Ours+Dropout also retains random-pruning accuracy close to that of Dropout, although its unpruned accuracy is lower than that of Dropout for most budgets. These results suggest that stochastic masking and margin-based robust training address complementary aspects of pruning robustness.

Nevertheless, these empirical results do not constitute formal guarantees. Dropout alone provides no certificate against worst-case pruning, whereas our framework can certify robustness over all admissible pruning masks within a given sparsity budget.

