# OpenReview forum: "Verification and Training of Neural Networks for Robustness Against Neuron Pruning"
_TMLR — Under review for TMLR_

### Review · Reviewer_2N4q · 2026-06-15

**Summary Of Contributions:**

The authors study fully connected feedforward neural network pruning from an adversarial perspective. In the selected setup, an adversary can remove contributions from an agreed-upon fraction of neurons at each layer, and the goal is to: (i) estimate the worst-case accuracy for a given input, and (ii) devise a training strategy that can alleviate the accuracy loss incurred by adversarial pruning.

For problem (i), the authors reformulate the problem as a mixed-integer linear program that can be approached with standard solvers, and for problem (ii), they derive an efficient upper bound on the logit margin that can be used as a training objective.

The proposed constructions are evaluated on an extensive set of experiments covering various adversarial scenarios and baselines.

**Audience:**

Yes

**Audience Explanation:**

The research lies at the intersection of several disciplines, including model compression, adversarial robustness, and certified robustness; therefore, a wide audience could potentially be interested in the presented findings.

**Claims And Evidence:**

Yes

**Claims Explanation:**

The main claims are well-supported by numerical experiments (including various ablations and comparisons with baselines) and theory.

**Requested Changes:**

1. (Section 1, Introduction) I understand the general formulation of the problem and do not dispute the theoretical and empirical results that the authors presented. However, I find the setup itself a little bit artificial. In the introduction, the problem is motivated primarily by hardware failures or constraints (e.g., "hardware faults can effectively silence neurons during inference", "SIMD-aligned or block-aligned pruning patterns that may not coincide with the mask chosen by a given algorithm"). Can the authors clarify why robustness to adversarial pruning is needed for such cases?

2. (Section 3, Problem formulation) "These are the only norms used throughout the analysis." The authors define only $L\_2$ norms but later use an $L\_0$ norm in Section 3.1, which is left undefined.

3. (Section 3, Problem formulation) I suggest specifying whether the output is assumed to be transformed by a softmax function, meaning that the outputs of the network are logits.

4. (Section 3.2, Verification and Training for Pruning Robustness) Equation (6) is described as follows: "The objective is to train the network parameters so that this deviation remains small relative to the nominal margin $\gamma(x, t) = f^{(t)}(x) - \max_{i\neq t} f^{(i)}(x)$, which ensures that the prediction is preserved under all admissible pruning masks." This is not what the loss function seems to encourage, because the second term is not literally the nominal margin, but a related quantity. I suggest either reformulating this sentence or clarifying how $\gamma(x, t)$ is related to the second term in Equation (6).

5. (Section 4.1, Mixed-Integer Linear Programming for Pruning-Robust Verification) The section reformulates the problem as an MILP in Equation (10). The formulation is exactly the same as Equation (5), so it may be unclear to a reader unfamiliar with the referenced literature why one even needs all the inequalities the authors present at the beginning of the section. The role of these inequalities is clarified in Remark 2, but this comes at the very end of the section. I suggest that the authors restructure the text slightly to put the clarification of why the inequalities are needed before introducing them.

6. (Section 4.2, Upper bound on the worst-case margin deviation for robust training objective) Both proofs for Proposition 1 and Proposition 2 are simple, clear, and already given in the text in sufficient detail, yet the authors reference the Appendix for them. The additional extended proofs in the Appendix seem to be redundant.

7. (Section 5.2, Robustness under Adversarial Pruning) "able 1 shows that our method" -> "Table 1 shows that our method"

8. (Section 5.4, Analysis under Different Pruning Methods) From the results presented in Table 6, it seems that TD performance is very competitive on non-adversarial and non-random pruning methods. As I understand it, TD was designed specifically for these use cases. I find it interesting that TD results in worse performance than standard training on random pruning. It seems that the original dropout method, which zeros out random neurons, should be more robust to random pruning. Did the authors try applying a method closer to original dropout to specifically target random pruning?

---

> ### Author Response · Authors · 2026-07-18
> **Official Comment by Authors**
>
> We thank the reviewer for the assessment of the paper and the constructive suggestions. We address the raised questions and concerns accordingly.
>
> **1. In the introduction, the problem is motivated primarily by hardware failures or constraints (e.g., "hardware faults can effectively silence neurons during inference", "SIMD-aligned or block-aligned pruning patterns that may not coincide with the mask chosen by a given algorithm"). Can the authors clarify why robustness to adversarial pruning is needed for such cases?**
>
> We thank the reviewer for raising this point. We agree that the original motivation overstated the direct connection to hardware failures and deployment constraints. Our goal is not to model these phenomena themselves, nor to assume that an adversary literally controls the pruning process. Rather, we use adversarial pruning as a conservative worst-case abstraction of uncertainty in the neuron-removal pattern.
>
> Indeed, in many deployment settings, the realized pruning pattern is not fixed in advance. For example, dynamic inference methods may select different subnetworks for different inputs or resource budgets [1, 2]. Heterogeneous federated or on-device systems may deploy different pruning masks and sparsity levels across clients or devices [3].
>
> In such settings, robustness to a single pruning scheme may be insufficient, as it requires a new verification procedure and new analyses every time the mask-selection mechanism changes. Our formulation therefore targets a stronger *mask-agnostic guarantee* that remains valid for any pruning policy whose resulting mask lies within the prescribed budget. We acknowledge that this generality comes at the cost of a stronger and potentially more conservative robustness requirement.
>
> We have revised the Introduction to emphasize this motivation better and clarify the use of adversarial pruning as a conservative worst-case.
>
> **2. The authors define only l_2  norms but later use an l_0 norm in Section 3.1, which is left undefined**
>
> We have revised the notation to explicitly define the $\ell_0$ quantity used in the sparsity constraints: $\||\cdot\||_0$ to denote the cardinality of a vector's support.
>
> **3. I suggest specifying whether the output is assumed to be transformed by a softmax function, meaning that the outputs of the network are logits**
>
> We have clarified that f(x) denotes the pre-softmax logit vector and that prediction is based on its argmax. Since softmax preserves the ordering of logits, it is omitted from the analysis.
>
> **4. Equation (6) is described as follows: "The objective is to train the network parameters so that this deviation remains small relative to the nominal margin γ(x,t), which ensures that the prediction is preserved under all admissible pruning masks." This is not what the loss function seems to encourage, because the second term is not literally the nominal margin γ(x,t), but a related quantity**
>
> Equation (6) measures the worst-case deviation of any pairwise class margin under admissible pruning masks. Let $\gamma(\mathbf{x},t,i)$ and $\hat{\gamma}_{m}(\mathbf{x},t,i)$ denote the pairwise margins between the true class $t$ and class $i \neq t$ before and after pruning, respectively.
>
> Since every pairwise margin can change by at most $\Delta(S,\mathbf{x},t)$, we have
>
> $$
> \hat{\gamma}_{m}(\mathbf{x},t,i)
> \ge
> \gamma(\mathbf{x},t,i)-\Delta(S,\mathbf{x},t).
> $$
>
> Therefore, if $\Delta(S,\mathbf{x},t)<\gamma(\mathbf{x},t)$, then $\Delta(S,\mathbf{x},t)<\gamma(\mathbf{x},t,i)$ then $\hat{\gamma}_{\mathbf m}(\mathbf{x},t,i)>0$ for every class $i\neq t$. It follows that, for every admissible pruning mask, all true-class pairwise margins remain positive and the predicted class is unchanged. This motivates training objectives that encourage the worst-case margin deviation $\Delta(S,\mathbf{x},t)$ to remain small relative to the nominal margin $\gamma(\mathbf{x},t)$.
>
> We have clarified this relation in the revised manuscript; please refer to **Problem 2** and the accompanying discussion.
>
> **5.  I suggest that the authors restructure the text slightly to put the clarification of why the inequalities (7)-(8)-(9) are needed to reformulate the problem as an MILP in Equation (10) before introducing them**
>
> We appreciate this suggestion and have restructured **Section 4.1** in the revised version so that the role of the MILP constraints is explained before introducing the formulation.
>
> **6. Both proofs for Proposition 1 and Proposition 2 are simple, clear, and already given in the text in sufficient detail, yet the authors reference the Appendix for them. The additional extended proofs in the Appendix seem to be redundant**
>
> We agree that some of the extended proofs are redundant given the level of detail already provided in the main text. We streamlined the appendix accordingly.
>
> **7. Typo: "able 1"**
>
> We fixed it in the revised version

---

> > ### Author Response · Authors · 2026-07-18
> > **Official Comment by Authors (continued)**
> >
> > **8. It seems that the original dropout method, which zeros out random neurons, should be more robust to random pruning. Did the authors try applying a method closer to original dropout to specifically target random pruning?**
> >
> > We thank the reviewer for this insightful suggestion. Motivated by this comment, we conducted additional experiments that evaluated three training strategies: (i) standard Dropout, which applies stochastic neuron masking during training; (ii) our margin-based robust training objective (Ours); and (iii) the combination of the two (Ours+Dropout).
> >
> > The additional experiments support the reviewer's intuition that standard Dropout is a strong baseline for random pruning. Moreover, combining Dropout with our robust training objective substantially improves robustness to adversarially optimized pruning masks found by PGD, while largely retaining the resilience to random pruning provided by Dropout. The improvement is particularly pronounced under multi-layer pruning budgets. These results suggest that stochastic neuron masking and our training objective provide complementary forms of pruning robustness.
> >
> > Full results and discussion are provided in **Section A.8** in the Appendix.
> >
> >
> >
> > *References:*
> >
> > [1]  Conditional computation in neural networks for faster models, ICLR 2016
> >
> > [2]  Runtime neural pruning, NIPS 2017
> >
> > [3]  Adaptive model pruning-expanding for federated learning on mobile devices. IEEE Transactions on Mobile Computing 2024

---

### Review · Reviewer_fuXK · 2026-06-20

**Summary Of Contributions:**

To my understanding, the primary contributions of this paper are threefold:
1. The authors formulate structured neuron pruning in fully connected layers as a worst-case verification problem, requiring the model to maintain its prediction under all pruning masks that satisfy a layer-wise sparsity budget .
2. They propose a Mixed-Integer Linear Programming (MILP) method to verify whether the worst-case pairwise margin remains positive for a given input and pruning budget.
3. They derive computable upper bounds on the margin deviation utilizing spectral norms and top-k activations, which are subsequently used to design a margin-aware, pruning-robust training objective.

**Additional Comments:**

no

**Audience:**

Yes

**Audience Explanation:**

Structured pruning, model compression, neural network verification, and certified robustness are all topics of significant interest within the machine learning community . Providing a worst-case robustness guarantee over all pruning masks satisfying a sparsity budget is a highly relevant research direction.

**Claims And Evidence:**

Yes

**Claims Explanation:**

The foundational proof strategies for Propositions 1–4 present no obvious mathematical errors . These proofs logically apply the Cauchy-Schwarz inequality, the triangle inequality, the 1-Lipschitz property of the activation function, and spectral norm error propagation. However, I'm not very familiar with this specific field, in general, I think the main proof of the article can hold true, but there are some places in the current version that are prone to mislead readers about the relationship among analytical bound, MILP certificate and training surrogate loss.

1. The recursive definition of the pruned network in Section 3.1 requires unification. Equation 2 defines the pre-activation as $\hat z_k = W_k \hat x_{k-1}$ rather than using the pruned $\hat x_{k-1}^p$. I view this as a notation issue rather than a fundamental modeling flaw, but since the formalization of Problem 1 relies on this recursion, the notation must be corrected and unified throughout the text.
2. Equation 6 defines the worst-case margin deviation using an absolute value: $\left|(\hat f_m^{(t)}(x)-\hat f_m^{(i)}(x))-(f^{(t)}(x)-f^{(i)}(x))\right|$, which successfully bounds the total magnitude of margin. However, from a robustness certification standpoint, the actual risk is a decrease in the margin, not an increase. The Cauchy-Schwarz derivation in Proposition 1 directly yields the lower bound $\hat f_m^{(t)}(x)-\hat f_m^{(i)}(x) \geq f^{(t)}(x)-f^{(i)}(x) - |W_{K+1}[t]-W_{K+1}[i]|_2|\Delta x_K|_2$ without requiring the absolute value. Wrapping the deviation in an absolute value is a highly conservative choice that unnecessarily penalizes benign pruning masks that increase the margin (i.e., improve prediction confidence), treating them as deviations to be minimized. The authors should justify this design choice or discuss whether it renders the training objective overly restrictive, thereby limiting model capacity.
3. The training objective proposed in Section 4.3 is a practical, bound-guided surrogate loss , but the associated claims should be adjusted. The sufficient condition utilized is $\Delta(S,x,t)\leq \gamma(x,t),$ which guarantees the prediction remains unchanged after pruning. Minimizing the proposed loss does not automatically guarantee that every sample will satisfy this condition post-training. The authors should explicitly distinguish between "the training objective encourages pruning robustness" and "post-training verification certifies pruning robustness". Claiming certified robustness requires an independent post-hoc check of $\Delta(S,x,t)\leq \gamma(x,t)$ or the MILP verification.
4. Some minor notation and writing issues should also be corrected. For instance, $\Delta x_k$ should explicitly depend on the pruning mask, e.g., $\Delta x_k(m)$. The notation $M$ and $M_{\mathrm{pruned}}$ should be used consistently. In Appendix A.1, “Preposition” should be “Proposition”, and “beacause” should be “because”. The phrase “Details of the proof are provide” should be corrected to “Details of the proof are provided”. The lower/upper bound notation in the MILP section should also be made clearer.

**Requested Changes:**

As mentioned above, the core theoretical contributions of this paper are valid. Therefore, I recommend acceptance if the authors addressing these minor revisions and clarifications.

---

> ### Author Response · Authors · 2026-07-19
> **Official Comment by Authors**
>
> We thank the reviewer for their input and helpful comments. Now we will address the concerns raised below.
>
> **1.  Equation 2 incorrectly defines the pre-activation**
>
> The reviewer is correct that the pre-activation should instead be defined using the pruned activation from the previous layer. We have corrected Equation (2) in the revised manuscript.
>
> **2. Wrapping the deviation in an absolute value is a highly conservative choice that unnecessarily penalizes benign pruning masks that increase the margin (i.e., improve prediction confidence), treating them as deviations to be minimized**
>
> Thank you for this important observation. We agree that only decreases in the pairwise margins can compromise robustness certification. Our use of the absolute value therefore defines a stronger, two-sided notion of pruning-induced margin deviation, yielding a sound but potentially conservative sufficient condition because margin increases are also treated as deviations.
>
> We adopt this symmetric formulation because it provides a uniform measure of **margin sensitivity** and leads directly to the tractable norm-based bound used in training. We have clarified this design choice and its conservatism in **Remark 2** and identify tighter one-sided bounds as future work.
>
> **3. The authors should explicitly distinguish between "the training objective encourages pruning robustness" and "post-training verification certifies pruning robustness"**
>
> We have revised **Remark 5** to state explicitly that the proposed loss only encourages the sufficient condition $\delta(S,\mathbf{x},t)\leq\gamma(\mathbf{x},t)$ during training and does not itself provide a certification guarantee. Pruning robustness for a given input must instead be established through an independent post-hoc verification step.
>
> **4. Some minor notation and writing issues**
>
> Thank you for catching these issues. We have fixed them in the revised version.
>
> We truly appreciate the reviewer's feedback and hope these clarifications and updates have addressed their concerns.

---

### Review · Reviewer_iDJv · 2026-07-08

**Summary Of Contributions:**

This paper studies robustness of neural networks against structured neuron pruning. The authors formulate neuron pruning as binary masks over hidden neurons (which forms an exponentially large set), and define pruning robustness as preserving the prediction under all masks satisfying a budget. They propose a MILP-based verification method for this robustness notion. They also derive an upper bound on the worst-case change of pairwise class margins under pruning, and use this bound to design a margin-aware robust training objective. Experiments are conducted on MNIST and CIFAR-10 to compare verified accuracy and empirical robustness under different pruning budgets and pruning methods.

**Audience:**

Yes

**Audience Explanation:**

Pruning is certainly a classical and important machine learning problem, and robustness to pruning is also a meaningful question. The TMLR audience would be certainly interested in it. However, on the other hand, I mainly have the following concerns:

1. In terms of experiments: The experiments that this paper do is solving pruning for a four layer neural network on MNIST/CICAR 10. This looks too"classical", and a bit weak for a paper published in 2026. I am not sure people working on empirical pruning would find these results very informative, as modern pruning settings usually involve larger architectures and more realistic structured pruning, such as channel, filter, or attention-head pruning.

2. In terms of the theory: The proof provided by this paper is sound and looks correct. However, they also seems to be a bit fairly standard. I take the proof 2 as an example: only two things are used here: one is the Lipschizness of simga, which linearize the problem, drops the non-linearitly, essentailly making the model a linear NN; and Cauchy-Shwarz inequality, which is  standard norm inequality.  This gives a valid upper bound, but I am not sure it provides much new insight into the pruning problem. One can certainly bound the error of a linear NN by expanding the norm layer-by-layer. The verification part looks like a natural application of existing MILP-based neural network verification to pruning masks.

Finally, I have doubts about how practical the robustness notion in this paper is. My understanding is that the verification problem certifies a worst-case property over all pruning masks satisfying a budget. Therefore, if the verification succeeds, then every admissible pruning mask in this (exponentially large!) set preserves the prediction for the verified input. This is a strong guarantee, but I am not sure how useful it is for practical pruning. In real pruning applications, one usually wants to find a good sparse subnetwork or a good structured pruning pattern, rather than require that every possible pruning pattern under the same budget works. Since the verification is against the worst-case mask, the guarantee may be overly pessimistic.

**Claims And Evidence:**

Yes

**Claims Explanation:**

I have went through the proof given in this paper, and I can fairly say that they are correct.

**Requested Changes:**

In terms of theory, based on the discussion above, my understanding is that the paper essentially uses the Lipschitzness of (\sigma) to reduce the nonlinear network to a layer-by-layer norm propagation argument, and then bounds the error through this recursive expansion. I believe this bound is likely to be loose and may not provide very meaningful insight into the pruning problem.

In terms of experiments, it would be useful to see whether the proposed method can be applied to larger-scale models and more realistic pruning settings. I would also like to see a more detailed discussion of the computational cost of the proposed training loss, especially how efficient it is to compute the loss and its gradients during training.

---

> ### Author Response · Authors · 2026-07-19
> **Official Comment by Authors**
>
> We thank the reviewer for their input and insightful comments. We will now address the questions raised below.
>
> **1. The experimental evaluation focuses on relatively classical settings (a four-layer neural network on MNIST/CIFAR10). It is unclear how informative the results are for modern pruning scenarios involving larger architectures and structured pruning such as channel, filter, or attention-head pruning.**
>
> We agree that experiments on channel, filter, or attention-head pruning and larger-scale architectures would provide a stronger evaluation of the applicability of the proposed certification framework. However, we would also like to clarify that the current evaluation is not restricted to a four-layer network; it includes a convolutional CIFAR-10 model and VGG11, although pruning is currently applied only to their fully connected layers. Across these settings, we demonstrate improvements in both certified and empirical robustness.
>
> Our work specifically targets formal guarantees under uncertainty in the neuron-removal pattern, which are relevant when empirical evaluation of a selected pruning mask alone is insufficient. At the same time, certifying robustness against all admissible pruning masks requires reasoning over a combinatorial mask space, which poses a significant scalability challenge and motivates the tractable bound developed in this work. We acknowledge that extending the framework and its evaluation to larger architectures and other forms of structured pruning remains an important direction for future work.
>
> **2. The theoretical analysis appears sound but relies largely on standard tools. Lipschitzness of sigma essentially makes the model a linear NN. The bound does not provide much new insight into the pruning problem. The verification part looks like a natural application of existing MILP-based verification.**
>
> We agree that the individual mathematical tools used in the analysis, including Lipschitz continuity and the Cauchy-Schwarz inequality, are standard. Our contribution is to specialize them to the combinatorial problem of structured neuron pruning and derive a tractable certification and training framework over all pruning masks that satisfy layer-wise sparsity budgets.
>
> We would also like to clarify that the Lipschitz argument **does not** linearize the network or reduce it to a linear neural network. *The nonlinear activation functions remain part of the network. Pruning-induced deviations still propagate through nonlinear activation; Lipschitz continuity is used only to upper-bound their magnitude.* The resulting bound does not assume a fixed activation pattern and therefore remains valid even when pruning changes the nonlinear activation states throughout the network.
>
> The pruning-specific part of the analysis lies in decomposing the layer-wise deviation into two mechanisms: (i) the local worst-case error induced by removing a budgeted subset of neurons and (ii) the propagation of deviations accumulated from preceding pruned layers. Propositions 3 and 4 presented this intuition.
>
> Similarly, we agree that the MILP formulation builds on established neural-network verification techniques. Its contribution lies in jointly encoding the discrete pruning masks, layer-wise sparsity constraints, neuron-gating operations, and network activation states so that the worst-case pruning configuration can be optimized over without enumerating the exponentially large mask space.
>
> **3. How practical the robustness notion in this paper is. Since the verification is against the worst-case mask, the guarantee may be overly pessimistic.**
>
> We agree that conventional static pruning typically aims to identify a single high-performing sparse subnetwork, as the reviewer notes. In safety-critical settings, however, empirical performance of a selected pruning mask may be insufficient, and one may additionally require a formal guarantee that a given pruning scheme preserves model correctness under the pruning patterns it produces.
>
> Such guarantees are pruning-scheme specific and do not automatically transfer when the pruning criterion, mask-selection policy, or deployment configuration changes, requiring the guarantee to be re-established for the new setting. Indeed, in some deployment scenarios, the realized pruning pattern is not fixed in advance. For example, dynamic inference methods may select different subnetworks for different inputs or resource budgets [1, 2]. Heterogeneous federated or on-device systems may deploy different pruning masks and sparsity levels across clients or devices [3].
>
> Our formulation, therefore, targets a stronger mask-agnostic guarantee that remains valid for any pruning policy whose resulting mask lies within the prescribed budget. This avoids re-establishing a scheme-specific guarantee whenever the mask-selection mechanism changes. We acknowledge that this generality comes at the cost of a stronger and potentially more conservative robustness requirement.

---

> ### Author Response · Authors · 2026-07-19
> **Official Comment by Authors (continued)**
>
> **4. I believe the margin deviation bound is likely to be loose and may not provide very meaningful insight into the pruning problem.**
>
> We agree that recursive Lipschitz- and norm-based propagation can introduce conservatism, as discussed in the manuscript. However, the role of the proposed bound is **not limited to providing a stand-alone verification certificate**. It is designed to be computationally *tractable and piecewise differentiable*, allowing it to serve as a **practical training signal** for improving pruning robustness.
>
> Empirically, as shown in Figure 4, the bound provides **non-vacuous certificates** over a meaningful range of pruning budgets. More importantly, incorporating the bound into the training objective **improves both certified and empirical robustness** over standard training in our experiments. These results suggest that, despite its conservatism, the bound captures pruning-sensitive information that is useful for both verification and robust training.
>
> **5. It would be useful to see whether the proposed method can be applied to larger-scale models and more realistic pruning settings. I would also like to see a more detailed discussion of the computational cost of the proposed training loss, especially how efficient it is to compute the loss and its gradients during training.**
>
> We agree that extending the framework beyond fully connected neuron pruning to larger architectures and other structured pruning settings is an important direction for future work. As noted above, our current evaluation also includes SmallConv and VGG11 on CIFAR-10, although pruning is currently applied only to their fully connected layers.
>
> Regarding computational cost, **Remark 4** provides a detailed complexity analysis of the bound computation, including spectral-norm estimation and the local pruning terms. **Table 4** further reports the end-to-end empirical training overhead, including both bound computation and gradient backpropagation.
>
> We sincerely thank the reviewer for their constructive feedback and hope that these clarifications address their concerns.
>
>
> References:
>
> [1] Conditional computation in neural networks for faster models, ICLR 2016
>
> [2] Runtime neural pruning, NIPS 2017
>
> [3] Adaptive model pruning-expanding for federated learning on mobile devices. IEEE Transactions on Mobile Computing 2024

---

### Author Response · Authors · 2026-07-19
**Official Comment by Authors**

We sincerely thank all reviewers for their valuable feedback and constructive comments. We are encouraged that all reviewers found the main claims to be well supported and the core theoretical development to be sound. We particularly appreciate Reviewer 2N4q's observation that “the main claims are well-supported by numerical experiments (including various ablations and comparisons with baselines) and theory.”

We have uploaded a revised manuscript addressing the comments raised during the review process. The main revisions include:

1. Clarifying the motivation, practical scope, and generality-conservatism trade-off of mask-agnostic pruning robustness in the Introduction.
2. Clarifying the margin-based condition in Problem 2, the distinction between robust training and post-hoc certification in Remark 5, and the conservatism introduced by the use of absolute margin deviation in Remark 2.
3. Restructuring the MILP formulation and clarifying the role of the activation and pruning constraints in Section 4.1.
4. Adding new experiments comparing our method with standard Dropout and with its combination with our robust training objective in Appendix A.8.
5. Strengthening the PGD-based pruning evaluation using a broader range of learning rates and random seeds. The corresponding values in Tables 1 and 6 have been updated, while the relative ordering of the methods and the main conclusions remain unchanged.

We have also corrected the notation and presentation issues identified by the reviewers throughout the manuscript. All changes are highlighted in blue in the revised manuscript. Detailed responses to each reviewer are provided below.